



# Reliability of meteorological drought indices for predicting soil moisture droughts

**Devanmini Halwatura[1], Neil McIntyre[1], Alex M. Lechner[1], Sven Arnold[1]**

[1] Centre for Water in the Minerals Industry, Sustainable Mineral Institute, The University of Queensland, Australia

*Correspondence to*: D. Halwatura (d.halwatura@uq.edu.au)

**Abstract.** Meteorological drought indices based on precipitation and/or evaporation are commonly used to detect the presence, severity and duration of soil moisture droughts. However, it is debatable whether droughts can be adequately characterised using only precipitation and/evaporation, or whether more physical based methods using soil water deficits and pressures is necessary. To address this question, the performances of two commonly used meteorological drought indices,

the Standard Precipitation Index (SPI) and the Reconnaissance Drought Index (RDI), are evaluated against soil moisture droughts identified using a physically based soil water model.

Our analysis is based on three sites in Eastern Australia, each representing specific soil-climate conditions. Drought duration and severity were estimated using SPI and RDI and soil water pressure data were simulated with Hydrus-1D. The performance of the two drought indices was measured in terms of their correlation with simulated monthly minimum soil

water pressures, and their ability to estimate the frequency with which the simulated pressure drops below threshold values. There was a significant correlation between the two drought indices (SPI and RDI) and the monthly minimum soil water pressure. Failure rate (FR) and false alarm rate (FAR) of drought indices detect soil moisture drought reasonably well (FR and FAR is <50%) for both drought indices (SPI and RDI) and soil depths (5cm and 30 cm) (except Melbourne). Overall SPI performs better (except shallow soils in Bourke) than RDI. However an uncertainty of the FR and FAR in the soil water

retention curve is always higher than the uncertainties of drought indices. The complexity and the uncertainty in the model encourage to use the simple drought indices, however the model provide physically relevant soil water pressure values which are species specific for plants.

## 1. Introduction

Drought is one of the most complex, harmful and least understood type of climatic events, causing an annual average of 6–8 billion USD of damage globally (Keyantash and Dracup, 2002; Yagci et al., 2013; Saghafian et al., 2003; Edwards, 1997); and it is expected that the severity and frequency of droughts will increase in the future due to climate change (Heim Jr, 2002). Droughts are classified into meteorological, agricultural, hydrological, and socioeconomic droughts (Shiau et al.,

2012; Zargar et al., 2011; Passioura, 2007). Across these drought classes over 150 drought indices are available and widely accepted as measures for monitoring spatial and temporal variability of water shortage (Vicente-Serrano et al., 2010;

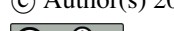



Quiring, 2009). Depending on the climatic region, type of drought and purpose of a study, drought indices will use information on rainfall, evaporation, soil moisture, surface water, groundwater or supply shortages (American Meteorological Society, 1997; Khedun et al., 2012; Hao and AghaKouchak, 2013). Often, measures of the severity and duration of droughts are derived from these indices, and drought frequency analysis is used to estimate the return period or

exceedance probability of a drought of given severity and/or duration (Halwatura et al., 2015a; She et al., 2016; Kwak et al., 2015).

Droughts play a critical role in agriculture and ecosystem restoration (Carrick and Krüger, 2007) - water is the key variable controlling many processes in the governing biogeochemical cycles (e.g., water, carbon, or nitrogen cycle). The movement of water from soil pores to plant roots and leaves follows potential energy gradients that are predominantly determined by

the soil moisture balance (Rodriguez-Iturbe, 2000; Prentice et al., 1992). If soil moisture is depleted, gradients in the soil water potential gradually break down, thereby affecting the vegetation negatively (e.g., reduction of crop production or failure of ecosystem establishment) (National Drought Mitigation Center, 2013). Agricultural drought indices such as Palmer Drought Severity Index (Palmer, 1965), Crop Moisture Index (Palmer, 1969), and Z index (Palmer, 1968) account for the water needs of plants and integrate soil moisture data. However, practical application of these indices is challenging as long-

term soil moisture data are rarely available and the establishment of soil moisture monitoring systems is expensive and time consuming. Simpler drought indices such as the Standard Precipitation Index (SPI) and Reconnaissance Drought Index (RDI) are often used when the necessary quality and quantity of observed soil moisture data are not available (Halwatura et al., 2015b).

Alternatively, physically based soil water models can be used to simulate soil water content, fluxes and pressures (Granier et

al., 1999; Hao and AghaKouchak, 2013; Anderson et al., 2012). Together with the local soil properties, long-term meteorological data (and potentially irrigation) can be used as inputs to estimate the probability of critical thresholds of soil water pressure (such as the wilting point) being exceeded. However, the application of physically based models for soil moisture drought estimation is challenging due to a range of issues such as uncertainty associated with the data and approximations of biophysical processes (Mishra and Singh, 2011; Chowdhary and Singh, 2010; Wang et al., 2011; Li et al.,

25 2005).

Therefore, in typical conditions where there are limited observations of soil moisture variables, the decision about how to estimate droughts related to soil moisture deficits requires the following questions be addressed: 1) Is it sufficient to use a simple drought index such as the SPI or RDI, and what threshold value should be employed to define a drought occurrence? 2) Compared to using a simple drought index, what value can a physically based soil moisture model provide considering its

uncertainty? The objective of this paper is to investigate these questions. The paper evaluates how successfully the SPI and the RDI can estimate the frequency of soil moisture deficits that have been synthesised using a physically based soil water model; and compares the success rates with those that may in practice be achieved by a soil water model considering its uncertainty. Our analysis is based on climate and soil hydraulic data from across three distinct sites in Eastern Australia.




Thorough out the paper we use the term 'soil moisture drought' instead of 'agricultural drought' because the research is relevant to a range of ecosystem restoration applications (Boken et al., 2005; Woli et al., 2012).

## 2. Methods

The method involves five steps (Fig. 1):

1. Selection of three sites representing typical climates and soil types in Eastern Australia.
2. Calculation of SPI and RDI drought indices using time series of observed climate data, and estimation of corresponding time series of soil water pressure using a physically based soil water model.
3. Performance evaluation of the drought indices against the soil water pressure data. This includes identifying thresholds that define drought occurrence; and the design, evaluation and analysis of metrics of characterising how well the indices represent the physically based soil water pressure.
4. Analysis of the sensitivity of simulated soil water pressure to uncertainty in the soil hydraulic parameters.
5. Analyse the effect of that uncertainty in the physically based soil water model affecting the comparison with the drought index

### 2.1. Step 1. Selection of three sites representing typical climates in Eastern Australia

We selected Cairns, Bourke and Melbourne, representing tropical, arid, and temperate climatic regions across Eastern Australia (Halwatura et al., 2015a; Köppen, 1936) (Table 1, Fig. 2). In Cairns and Melbourne the soils are predominantly sandy loams, whereas in Bourke sandy clay loam is predominant (Australian Soil Resource Information System, 2011) (Table 2). While soils are variable in these regions, only these dominant soil types were considered. Historic daily rainfall and potential evaporation point data (calculated by Penman-Monteith equation) were available for these sites from 1971 to 2013 (Table 1) with a coverage of at least 97% (Bureau of Meteorology, 2013).

### 2.2. Step 2. Estimation of drought indices and soil water pressure

This step generated data for the two drought indices, SPI and RDI, and the simulated soil water pressure data. The SPI and RDI were calculated using climate inputs averaged over one, three and twelve month time periods. However for the sake of simplicity we present only the results based on the three month time period. The SPI is derived by fitting the rainfall record to a probability distribution and transforming it into a normal distribution with zero mean and unit standard deviation. Rainfall conditions greater or smaller than average rainfall are represented by positive or negative values of SPI respectively (McKee et al., 1993). Following McKee et al. (1993), the three-monthly SPI is calculated for each site as,

$$SPI_i = \Phi^{-1}\left(F_R \frac{(r_{i-2}+r_{i-1}+r_i)}{3}\right) \tag{1}$$



where r is the rainfall, $F_R$ is the non-exceedance probability of the three-month average value, calculated by fitting a Gamma distribution using the method of moments, the subscript $i$ is the month number ranging from the third to the last month in the record, and $\Phi^{-1}$ is the inverse cumulative distribution function of the standard normal distribution.

The standardised three-monthly RDI (Tsakiris and Vangelis, 2005) is calculated for each site as:

$$RDI_i = \frac{\ln(\gamma_i) - \mu_{\ln(\gamma)}}{\sigma_{\ln(\gamma)}}, \tag{2a}$$

with

$$\gamma_i = \frac{P_{i-2} + P_{i-1} + P_i}{PET_{i-2} + PET_{i-1} + PET_i} \tag{2b}$$

where P and PET are rainfall and potential evapotranspiration respectively, $i$ is the month number ranging from the third to the last month in the record, and $\mu_{\ln(\gamma)}$ and $\sigma_{\ln(\gamma)}$ are the arithmetic mean and standard deviation of $\ln(\gamma)$ over all its values for that site.

For the derivation of soil water pressure using the physically based soil water model (in addition to the climate data) water pressure characteristics of the soils were provided by the Australian Soil Resource Information System (2011). This data included the soil moisture content and water pressure at saturation (0 hPa), field capacity (-100 to -330 hPa), permanent wilting point (-15,000 hPa) and air-dry conditions (-1,000,000 hPa). The van Genuchten water retention curve was fitted to these data using RETC 6.02 (Van Genuchten et al., 1991) to estimate the first-order mass transfer coefficient α and curve shape parameter $n$. These two parameters were used, together with the soil water content at saturation ($\theta_s$), the residual water content ($\theta_r$) and the saturated hydraulic conductivity ($K_s$) (Table 2), to simulate one-dimensional water flow in Hydrus-1D (Šimůnek et al., 2008) and to estimate daily soil water pressure ($\psi$) over two different soil depth (see below). The daily rainfall and potential evapotranspiration inputs were the same as those used to calculate the drought indices. The lower boundary condition was set as free drainage and the upper boundary condition was rainfall and potential evapotranspiration with surface runoff. Appendix A lists the settings used in Hydrus-1D.

The primary assumptions of the soil water modelling were that (i) the (Australian Soil Resource Information System, 2011) soil water characteristics are representative of the selected soil types, including incorporating local vertical and lateral heterogeneity, (ii) using the effective parameter values of the water retention curve, Richards' equation is an adequate approximation of the unimodal soil permeability, (iii) a free drainage lower boundary condition is an adequate approximation, (iv) the evaporative demand primarily determines the upper boundary condition, (v) sub-daily variations in soil water pressure are not significant, and the use of daily averaged input and output data is sufficient, and (vi) there is no irrigation so that drought is not influenced by groundwater or surface water storage. The simulated water pressures were taken to be accurate for the purpose of the comparisons made in Step 3, while in Step 4 we introduce an uncertainty analysis.



### 2.3. Step 3. Evaluation of the drought indices against minimum soil water pressure

For each site and each month, we extracted the minimum soil water pressure (hPa) averaged from the soil surface to 5 cm depth ($\widehat{\psi^5_{min,i}}$), and also from the soil surface to 30 cm depth ($\widehat{\psi^{30}_{min,i}}$). We then evaluate the correlations between $pF^5 =$

$\log10(-\widehat{\psi^5_{min,i}})$ and $pF^{30} = \log10(-\widehat{\psi^{30}_{min,i}})$, and $SPI_i$ and $RDI_i$. We also evaluated the correlations between $pF^5$ and $pF^{30}$ and $SPI_{i-1}$ and $RDI_{i-1}$ in case the three month averages defined in equation (1) and (2) more strongly influence the next month's minimum soil water pressure.

The failure rate (FR) and false alarm rate (FAR) were also calculated to quantify the success of the drought indices in detecting critical periods of soil moisture deficit based on the simulated low soil water pressure events (See Appendix B.1

for further explanation):

$$FR = \frac{\#E(a)}{\#E(a+b)} \qquad (3)$$

where $\#E(a)$ is the number of simulated low soil water pressure events not detected by the drought index, and $\#E(a+b)$ is the total number of simulated low soil water pressure events.

Likewise, the FAR was calculated as:

$$FAR = \frac{\#\,E(c)}{\#\,E(b+c)} \qquad (4)$$

where $\#E(c)$ is the number of drought events detected by the drought index not corresponding to periods of low simulated

soil water pressure, and $\#E(b+c)$ is the total number of droughts detected by the drought index. Low values of FR and FAR indicates the capability of the drought index to reliably detect periods of low soil water pressure.

For each site the threshold that determines a soil moisture drought event was selected by the percentile of all simulated $pF^5$ and $pF^{30}$ that minimizes the FR and FAR (Fig. 3). Ideally, the threshold would be physically based; however, commonly used physically based thresholds, such as the wilting point, do not necessarily coincide with stress levels of plant

communities (Arnold et al., 2014). We choose the 75[th] percentile to represents the upper limit of potential performance. At the 75[th] percentile all drought index values below zero were taken as drought events. Together with the $pF^5$ and $pF^{30}$ thresholds, this defines the segments a, b and c used in equation (3) and (4) and illustrated in Appendix B. However the FR and the FAR will be same when we use same threshold of both drought index and soil water pressure (in this study 75[th] percentile), because the a and c values in equation (3) and (4) will be same.



## 2.4. Step 4. Sensitivity analysis of simulated soil water pressure

In this step we explored the effect of uncertainties in the water retention curve parameters on the simulated soil water pressure. We assessed the local sensitivity of the simulated soil water pressure to uncertainty in $\theta_r$, $\theta_s$, $\alpha$, and $n$. The
normalised relative sensitivity S was calculated for each site as,

$$S = \frac{d\langle \widehat{\psi_{min}^k} \rangle / \langle \widehat{\psi_{min}^k} \rangle^0}{dP_{WRC} / P_{WRC}^0} \tag{5}$$

where $\langle \widehat{\psi_{min}^k} \rangle$ is the simulated monthly minimum soil water pressure, $P_{WRC}$ refers to one of the four parameters in the soil water retention curve, the superscript 0 refers to the initial reference values of $\langle \widehat{\psi_{min}^k} \rangle$ and $P_{WRC}$, and $d\langle \ \rangle$ refers to the difference between the perturbed and reference values. Each of the four parameters was varied in turn, between values
$dP_{WRC}/P^0{}_{WRC} = [-0.5, 0, +0.5]$, while other parameters were kept at default values. The simulated mean soil water pressure was considered sensitive to uncertainty in the soil hydraulic model if $S > 1$.

## 2.5.  Step 5. The effect of uncertainty on the relative value of the physically based soil water model

In the final step we conducted a local sensitivity which indicates how uncertainty in the hydraulic model parameters affects the modelled soil water pressure and the calculation of FR and FAR. Using this method we explored how well we would expect the hydraulic model to detect drought events given scenarios of hydraulic model uncertainty. We used simulated soil water pressure based on the default parameters $(\psi_{min}^k)$ of the water retention curves (Table 2) as a benchmark representing the 'true' soil water pressure response. This was compared against the simulated soil water pressure based on the hydraulic
model parameter perturbations that caused the highest and median values of $S$, which are highlighted in bold and italics in Appendix C. The representing relative sensitivity values of assessing parameters ($\theta r$, $\theta s$, $\alpha$) were very similar to each other (Appendix C) therefore the mean values were used to calculate FR*.

Then we calculated FR* and FAR*as:

$$FR^* = \frac{\#E(a^*)}{\#E(a^* + b^*)} \tag{6}$$

and

$$FAR^* = \frac{\#E(c^*)}{\#E(b^* + c^*)} \tag{7}$$

where $\#E(a^*)$ is the number of low soil water pressure events simulated with the default water retention curve but not simulated with the perturbed water retention curve, $\#E(a^* + b^*)$ is the total number of low soil pressure events simulated with the default water retention curve, $\#E(c^*)$ is the number of low soil pressure events simulated with the perturbed water retention curve but not simulated with the default water retention curve, and $\#E(b^* + c^*)$ is the total number of low soil





pressure events simulated with the perturbed water retention curve. The comparison of default and perturbed minimum pressures is shown schematically in Appendix B1.

If FR*>FR or FAR*>FAR, the assumed parameter uncertainty in the hydraulic model critically affects its relative ability to detect droughts, so that the simple drought index may be preferred over the more complex soil water model, even if FR or
FAR is high.

## 3. Results

Drought indices and pF values (derived from soil water model) showed reliable all sited. For the most extreme droughts,
both pF values and drought index values showed similar results for all three sites (except $pF^5$ for Melbourne and Cairns). Drought index values were greatest (most negative) in arid Bourke (-2.94), similarly the pF values for both soil profiles were also highest in Bourke ($pF^5 = 4.47$ and $pF^{30} = 3.38$) compared to other two sites (Fig. 4). Melbourne (SPI=-2.57, $pF^{30}$=2.41) and tropical Cairns (SPI=-2.13, $pF^{30}$= 2.38) has the most severe drought accordingly next to Bourke, while the trend changed for $pF^5$ where tropical Cairns ($pF^5$=3.88) has the most extreme drought than temperate Melbourne ($pF^5$=3.69).
Both drought indices were significantly correlated ($p <0.05$) with soil water pressure, but the strength of the correlation varied with the depth of the soil profile and was stronger for the deeper (30 cm) soil profiles (Fig. 5). The correlations were strongest for tropical Cairns ($R^2 = 0.69$) and arid Bourke ($R^2 = 0.51$) for SPI and RDI respectively. Correlations between soil water pressure values and drought index values for the previous month were lower than default values indicating no evidence of a lag between climate and soil moisture response that is not captured within equation (1) and (2).
SPI performed better than RDI in terms of FR and FAR across all locations and soil profiles The FR and FAR of SPI ranged between 19% and 42% and the values of RDI ranged between 36% and 68% (Table 3). For RDI, the FR and FAR values were always higher for the deeper soil profiles, while for SPI they were lower for the deeper soil profiles (except tropical Cairns where the FR and FAR were the same for both profile depths). In temperate Melbourne, FR and FAR values derived using RDI were higher than for other locations (58% and 68% for the two soil depths), reflecting the particular difficulty of
accounting for PET at this site, while tropical Cairns show the greatest improvement moving from RDI to SPI (e.g. FR and FAR reduced from 46% to 19% for the 30cm soil depth (Table 3)) implying particular benefit in omitting the PET terms in Cairns' highly variable climate.

The 75% threshold in SPI and RDI, used to define the occurrence of drought, was selected for consistency with the 75% threshold in simulated soil water deficits, rather than to optimise FR and FAR performance in detecting droughts, so the
results represent the upper limit of potential performance; however the performance was quite sensitive to the selected percentile (the implications of this will be discussed later). Therefore, it is interesting to look at how FR and FAR vary over a range of selected SPI and RDI threshold values, while maintaining the 75% threshold applied to the simulated soil water





deficits. For each location and soil depth, the FR and FAR values were the same (Fig. 3) (as they must be when using the 75 percentile to define the denominators of both equation (3) and (4)).

The variation of FR and FAR over a range of SPI threshold values for each site shown in Fig. 3. As expected, when the SPI threshold is increased so that only extreme values of SPI are classed as drought occurrences, the FR becomes high and FAR becomes low. For example, using the 95[th] percentile, the FAR values for Cairns and Melbourne reaches zero and the FR values reach 81% (Fig. 3). Although there may be particular circumstances when low FR or FAR rates may be the aim, the results confirm that a balanced performance is achieved by calibrating the actual number of drought events (in this case represented by the simulation) so that it is equal to the modelled number of drought events (in this case using SPI and RDI).

The results of the sensitivity analysis are shown in Appendix C. The simulated soil water pressure was most sensitive to uncertainty in parameters of the water retention curve in tropical Cairns, where the relative sensitivity ranged from 1.4 (when perturbing n) to 4.0 (when perturbing θs). In other words, the simulated soil water pressure rose by up to 4 % after changing the default parameter values by 1%. In contrast, the highest sensitivities across the other locations ranged from 0.3 in temperate Melbourne to 2.7 in arid Bourke.

The performance of the drought index and soil water model is presented in Fig. 6 by showing the differences between FR*-FR, (the values of FAR*-FAR are identical). The results in Fig. 6 show that for all locations the drought indices were more reliable than the simulation approach, more so for Cairns (Cairns shows the highest sensitivity to parameters in water retention curve (Appendix C)) and Bourke, and less so for Melbourne. Generally, the simulation model performed relatively well for shallow soils than the deeper soil profiles (Fig. 6).

## 4. Discussion

In this study we assessed the ability of selected meteorological drought indices to identify periods of soil water drought through a comparison with the physically based soil-water model Hydrus-1D. The assessment used soil hydraulic properties and 25 years of climate data from three locations in eastern Australia representing tropical, temperate and arid climates. The study analysed the correlations between the indices and simulated soil water pressures, and the frequency with which the indices detected soil water deficit thresholds being passed. The relative abilities of the indices and the physically based model, considering scenarios of uncertainty in the latter, were also assessed. The drought indices have variable success in detecting the soil water deficits. The frequency which the drought indices failed to detect periods of soil water deficit and the frequency with which they falsely detected periods of soil water deficit (FR and FAR respectively) are generally below 50% (in all cases except at Melbourne using RDI) and as low as 19% (at Cairns using SPI) (Table 3). Due to the uncertainties in the simulation model parameters, which represent the level of uncertainty what may occur in a practical application, the drought indices are considered generally more reliable (Fig. 6). This highlights the importance of precise estimation of the soil water retention curve. Although our results show a strong preference for using the simplest index, SPI, it is reasonable to



also conclude that, when and where adequately supported by data and modelling expertise, a physically based soil water model should be used in preference.

One of the major concerns of using meteorological drought indices is falsely predicting a soil moisture drought or over-estimating its frequency (Sheffield et al., 2012; Touchan et al., 2005; Törnros and Menzel, 2014). For an example, the

Palmer Drought Severity Index has overestimated global drought severity over the past 60 years (Sheffield et al., 2012). Also the tree growth assessments based on tree ring analysis showed that SPI overestimated the tree ring index values (Touchan et al., 2005). Our results were 'calibrated' to achieve a balance between over-estimation (high FAR values) and under-estimation (high FR values) (Fig. 3), yet at best achieved FR and FAR values of 19%. There are several potential reasons why the performance was not better, which include the simplicity of the indices used, the disparity of time-scales and the

potential errors in the simulated soil water data. These are expanded upon below.

Precipitation and potential evaporation are treated simplistically in the calculation of the SPI and RDI (Shiau, 2006). Equations (1) and (2) contain combinations of the climate variables that give convenient and intuitive indices of drought (Shiau and Modarres, 2009; Vangelis et al., 2013), but are not optimised to approximate well the non-linear and site-specific relationships that govern soil water deficits, in particular they neglect the soil type (Wang et al., 2015). Some soils will retain

water better than others, for an example sandy clay loam in Bourke may retain more water than sandy loam in Cairns or Melbourne (Wong et al., 2016), and affect the number, depth and timing of soil water deficits. As there are more than hundred possible drought indices, some of which incorporate additional variables beyond precipitation and potential evaporation, there is scope to explore alternatives (Mpelasoka et al., 2008). However, notwithstanding the limited scope of this paper, our results point to the simplest being the best. Similar results have been observed in North Carolina showing that

SPI is more representative of soil moisture variation (Sims and Raman, 2002).

The climate data are averaged over 3 months before input to the monthly SPI and RDI calculation (Eqs 1 and 2), whereas the simulation used a daily time-step. The performance was potentially limited by this discrepancy in time-scales. For example, for the shallow soil, there were cases where three simulated soil moisture minima were contained within one long drought event identified by the index, while in other cases the simulated event was too short to be recognised by the more dampened

response of the index. Results show a poorer correlation between the indices and soil water pressure if a 12-month averaging period is used instead of a 3-month period (result not shown here), with the exception of Melbourne using SPI where the 12-month period is marginally better. In cases, higher correlations have been observed using SPI with an averaging period less than 3 months especially for the top 10 cm of soil (Sims and Raman, 2002), although longer timescales have been recommended in general for drought analysis (Du et al., 2013). Contrasting conclusions about the optimal averaging period

may be explained by differences in climate, soil and water management contexts (Vicente-Serrano et al., 2012; Xu et al., 2012). A 12-month timescale may not capture the significance of the five to six-month wet and dry cycle such as in Cairns; and a three-month timescale may not capture the annual scale lag-times of groundwater-fed irrigation schemes. Thus it is a



matter of finding the suitable timescale for each specific context, and it may be helpful to present results for alternative time-scales (Vicente-Serrano et al., 2012).

Soil moisture depletion primarily limits plant establishment and seedling survival because, although well-established trees have access to stable water reserves through their deep roots, seedlings with shallow roots may not have the access to the

water even though deeper soil levels are saturated (Padilla and Pugnaire, 2007; Heim Jr, 2002; Lloret et al., 1999). In our study, soil depths (0-5 and 0-30 cm) were selected to represent the root depth of vegetation (e.g. 5 cm for seedlings and 30 cm for grown vegetation). Given the more rapid variations in soil water content experienced in the shallow soil, if drought indices are used as a planning tool for initial establishment of seeds, a shorter time scale than three months is likely to be preferred. Generally, the soil water model performed rather well for shallow soils than the deeper soil profiles. We presume

this is because it operates on a daily time-step as opposed to the 3-monthly averages used by the RDI and SPI and therefore, despite its uncertainty, performs relatively well for short-lived, near surface soil moisture deficits.

The performances of the indices reported in this paper are conditional on the accuracy of the soil model Hydrus-1D. Although an uncertainty analysis was performed, this only involved arbitrary one-at-a-time perturbations to soil hydraulic property parameters, and did not explicitly examine the sensitivity of performance to structural errors in Hydrus such as

effects of anisotropy and heterogeneity of soil properties, the spatial-temporal distribution of plant water uptake, and the applied lower and upper boundary conditions. These errors may influence how the performances varied between the SPI and RDI indices, between the two depths and over the three sites. A particularly interesting result was that the SPI index, which excludes the effect of PET, performed considerable better than the RDI index, which may have been due at least partly to the accuracy with which our application of Hydrus simulated evaporation (Fig. 5). A potential approach to extending the

uncertainty analysis would be a Monte Carlo based global sensitivity analysis, as well as exploring some of the structural uncertainties such as the single porosity, evaporation model and boundary condition assumptions (Le Vine et al., 2016).

Overall, SPI performed better than RDI, illustrating that in general the inclusion of PET in RDI confounds the prediction of drought for these sites, although this result may be affected by PET estimation and Hydrus model errors as discussed above. This implies that accounting for the potential evaporation is more important for the shallower soils, although this result is

conditional on how Hydrus distributes the evaporation losses over depth. For arid Bourke, there was less benefit using SPI rather than RDI (Fig. 5), which we speculate is due to a stronger and more linear influence of PET on the soil moisture minima at that site (Khalili et al., 2011; Asadi Zarch et al., 2015). The differences in the correlations shown in Fig. 5 are consistent with the differences seen in FR and FAR values (Table 3), with both showing that the best predictor of droughts is SPI in particular at Cairns.

To avoid the inevitable model errors, ideally measured soil water data would be used. However, long time-series of observations representing field-scale soil water conditions are rare, and typically limited to a few years at best (Sims and Raman, 2002). Remotely sensed soil moisture data are increasingly available, but are expensive, cover only the last two decades, and shallow soil layers have limited accuracy (Houser et al., 1998; Dorigo et al., 2010; Mishra et al., 2015).





Another benefit of the simulation approach is that it allows the depths of interest can be selected, which can be valuable as root depths are species specific (Canadell et al., 1999).

Table 4 summarises the comparison of the merits of drought indices versus physically based soil water models for soil water drought estimation. The SPI and RDI require only meteorological data, which are often freely available, although they are less often available for the recommended minimum period of 30 years (McKee et al., 1993), and accuracy may be poor if the location of interest is far from a rain gauge. Data preparation and estimation of the drought is easy to follow and not time consuming (McKee et al., 1993; Tsakiris et al., 2007). However, the drought index only gives the anomaly from the average value, rather than a value that is physically meaningful. For an example the same drought index value for two different sites will show different soil water pressures, e.g. for an SPI value of -2, the corresponding soil water pressure (hPa) is -37.32 for Cairns and -2381.7 for Bourke.

Another limitation of the approach taken here is the limited length of climate time series. We used historic rainfall and potential evaporation data for 25 years, which represents only a limited range of drought extremes, and ideally longer-term data would be used. The use of monthly soil water pressure minima provided 300 samples for the assessment of performance, however clearly these samples are not independent droughts and a more statistically robust assessment would be based on annual minima, if enough samples (long enough time series) could be obtained. In our study we used the dominant soil type of the location (Australian Soil Resource Information System, 2011). However a good representation of existing or historical soil types may not be appropriate, because in many soil management applications, including soil rehabilitation and agriculture, involve the soil being modified (Hamrin et al., 2001).

## 5. Conclusions

The study reveals that a simple drought index, SPI, which uses only monthly precipitation as an input, may out-perform a more sophisticated index, RDI, over a range of soil types and climates. This was based on comparing the failure rate (FR) and false alarm rate (FAR), measures of how reliably the indices detected simulated soil water drought events, at three sites in Australia. The FR values obtained using SPI ranged from 19% to 32% when averaging soil water pressure over the top 30cm of the soil column and averaging precipitation over 3 months, which was considered a satisfactory performance. The principal reasons for the relative success of SPI results were proposed to be the over-simple treatment of potential evaporation reducing the correlation between the RDI and soil water pressure values, as well as the dominant effect of precipitation for the climates and soils considered. The second part of the paper compared the SPI performance with that expected from the simulation model under plausible uncertainty in the model's key parameter values. Results showed that the simulation model is unlikely to more useful than the SPI for soil water drought estimation due to model uncertainty at sites where good observations of soil water do not exist. Potential advantages of the simulation approach, however, is the ability to define soil water droughts using physically meaningful thresholds where specific knowledge of root depth distributions and water demands exists.



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



Table 1: Climate indices for selected locations with focus on rainfall R (subscripts w and s denote winter and summer, respectively) and potential evaporation PET.

| Location | | Length of meteorological data (years) | Climate index | | Köppen-Geiger[c] climate classification |
|---|---|---|---|---|---|
| | | | R/PET[a] | $R_w/R_s$[b] | |
| **Bourke** | (-30.233 S, 145.917 E) | 1971-1996 (25) | 0.20 | 0.61 | Arid, steppe |
| **Cairns** | (-17.457 S, 145.992 E) | 1988-2013 (25) | 0.91 | 0.10 | Tropical, savannah |
| **Melbourne** | (-37.966 S, 144.553 E) | 1988-2013 (25) | 0.51 | 0.95 | Temperate, without dry season |

a – (UNEP, 1992), b – Based on average of three-months of rainfall during winter (June – August) and summer (December – February), c –(Peel et al., 2007)

Table 2: Soil type and water retention parameters of selected sites and the ranges used for the sensitivity analysis.

| Soil parameter | Cairns | Melbourne | Bourke |
|---|---|---|---|
| **Soil type** | Sandy loam | Sandy loam | Sandy clay loam |
| $\theta_r$(cm$^3$ cm$^{-3}$) | 0.05 (0.025-0.06) | 0.1(0.05-0.15) | 0.07 (0.035-0.105) |
| $\theta_s$(cm$^3$ cm$^{-3}$) | 0.47 (0.235-0.705) | 0.513 (0.025-0.769) | 0.33 (0.165-0.495) |
| $\alpha$ (cm$^{-1}$) | 0.1864 (0.093-0.279) | 0.1114 (0.055-0.167) | 0.023 (0.012-0.034) |
| n (-) | 1.2087 (1.087-1.813) | 1.3695 (1.232-2.05) | 1.296 (1.166-1.944) |
| $k_s$( cm day$^{-1}$) | 106.1 | 106.1 | 31.44 |

Table 3: Performance analysis of drought indices based on monthly minimum soil water for the threshold value of 75[th] percentile

| Location | Soil depth (cm) | Drought index | |
|---|---|---|---|
| | | FR (%) | |
| | | RDI | SPI |
| **Bourke** | 5 | 36 | 42 |
| | 30 | 41 | 32 |
| **Cairns** | 5 | 45 | 19 |
| | 30 | 46 | 19 |
| **Melbourne** | 5 | 58 | 41 |
| | 30 | 68 | 32 |

Note: FAR values are identical to FR values





Table 4: The pros and cons of the alternative approaches to soil moisture drought estimation

| | Drought indices | | Soil water pressure |
| | SPI | RDI | |
|---|---|---|---|
| **Data requirement** | rainfall | rainfall, evaporation | rainfall, irrigation, evaporation, soil type and soil hydraulic parameters |
| **Time for data/model preparation** | low | low | high |
| **Calculation time** | low | low | moderate |
| **Cost for data** | mostly free | sometimes have to purchase | may expensive for some locations |
| **Data availability** | available for most locations | available for some locations | restricted to few locations or have to measure in sites |
| **Applicability to any climatic region** | have issues with arid regions | can apply to any climate | can apply to any climate |





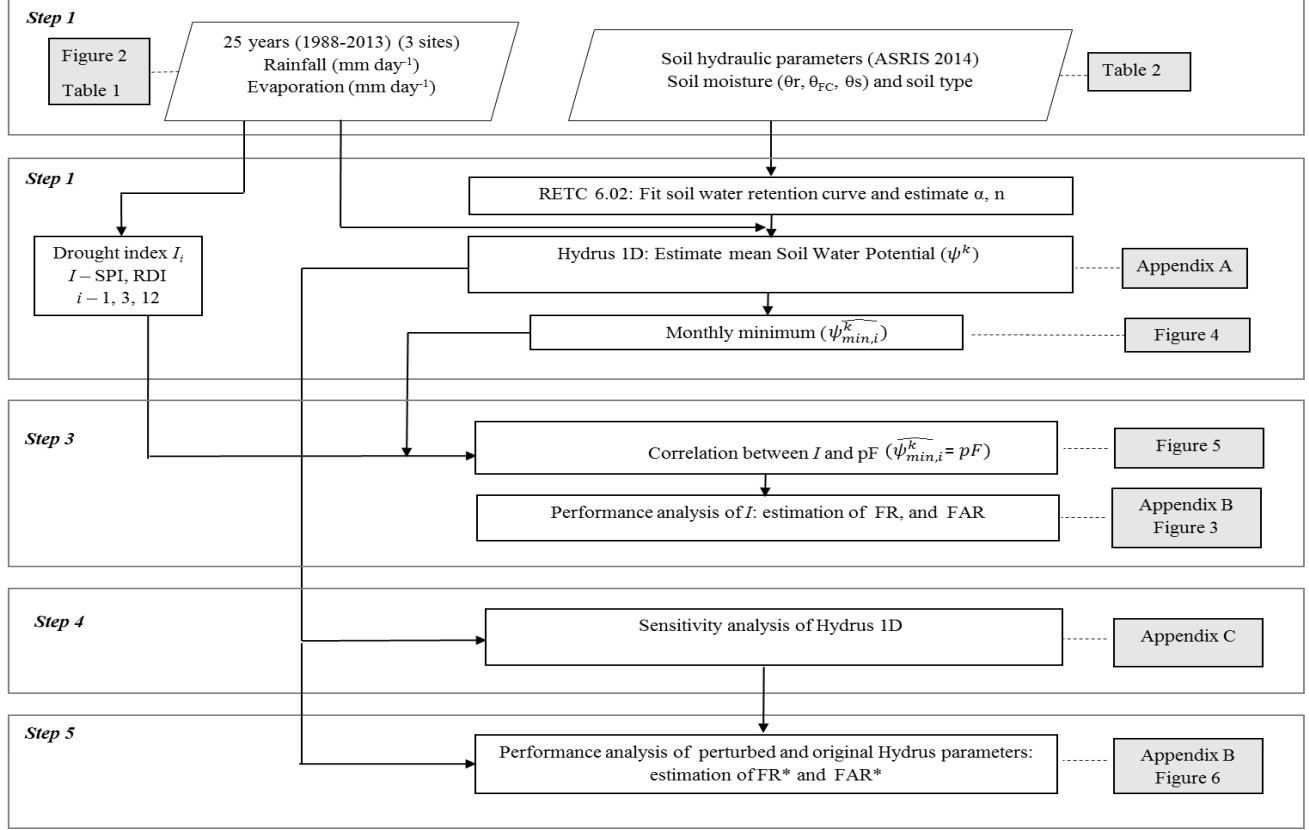

Figure 1: Schematic diagram of steps applied in the methodology. See Section 2 for further details. Step 1: Selection of three sites representing typical climates in Eastern Australia. Step 2: Estimation of drought indices and soil water pressure. Step 3: Evaluation of the drought indices against minimum soil water pressures. Step 4: Sensitivity analysis of simulated soil water pressure. Step 5: The effect of uncertainty on the relative value of the physically based soil water model.



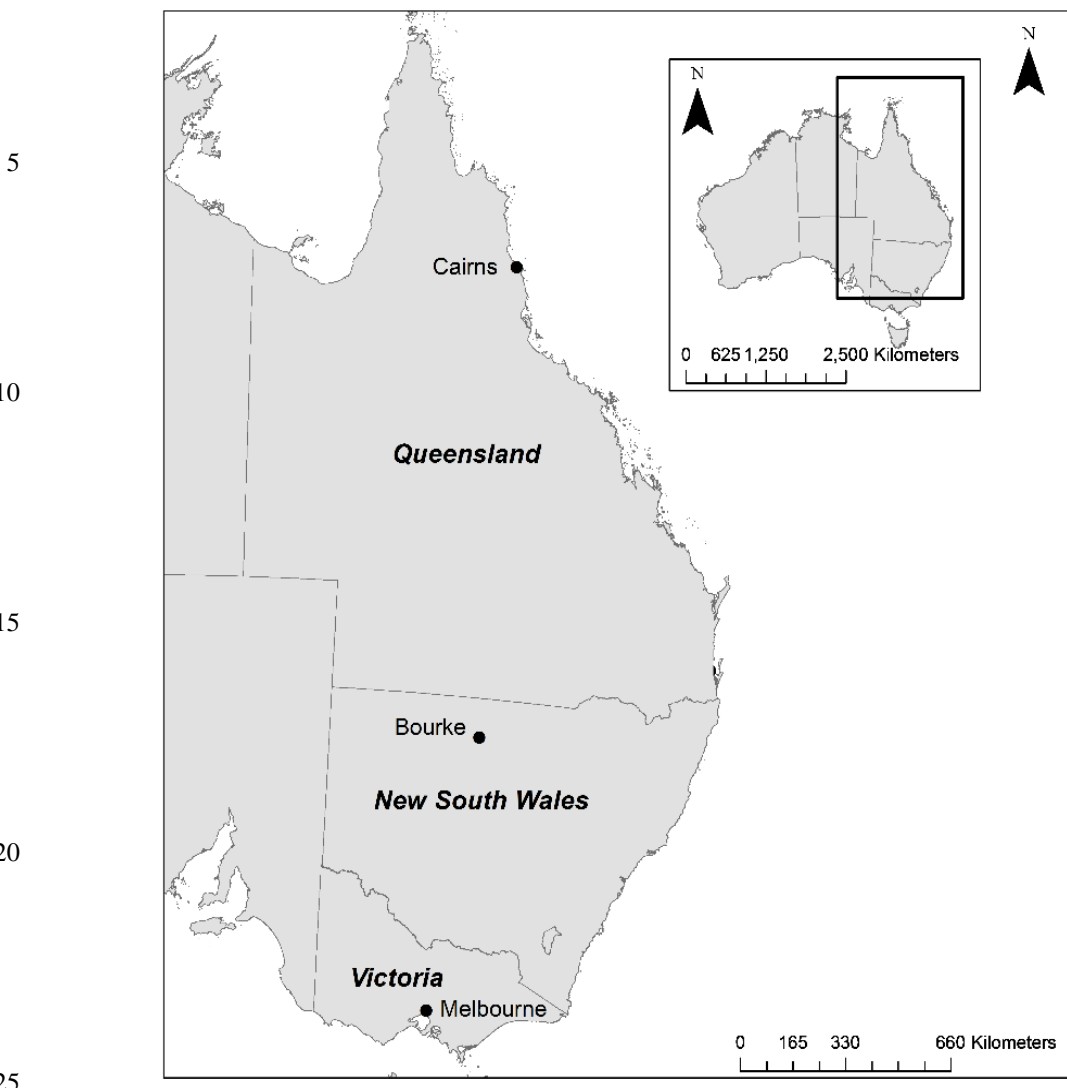

Figure 2: Selected locations of interest





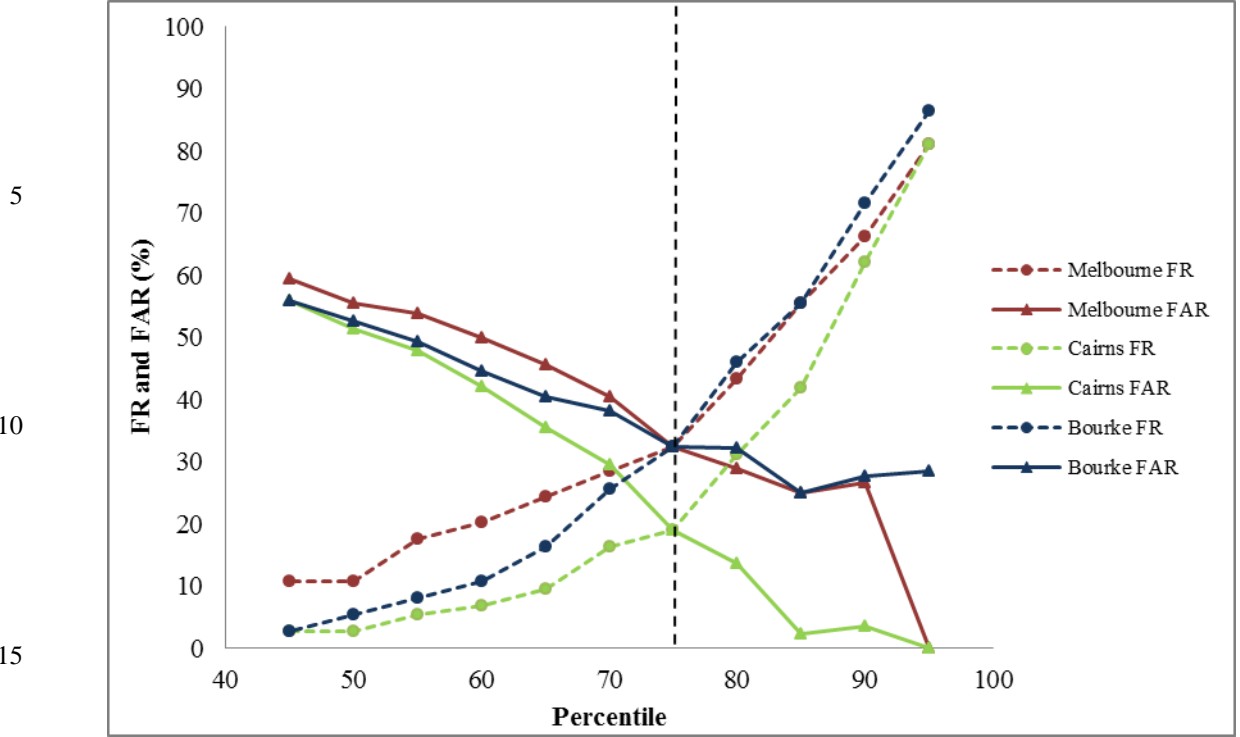

Figure 3: The change of FR and FAR with the increase of threshold percentile of SPI for the 30 cm spoil depth. Dotted line
shows the 75[th] percentile used in the study.




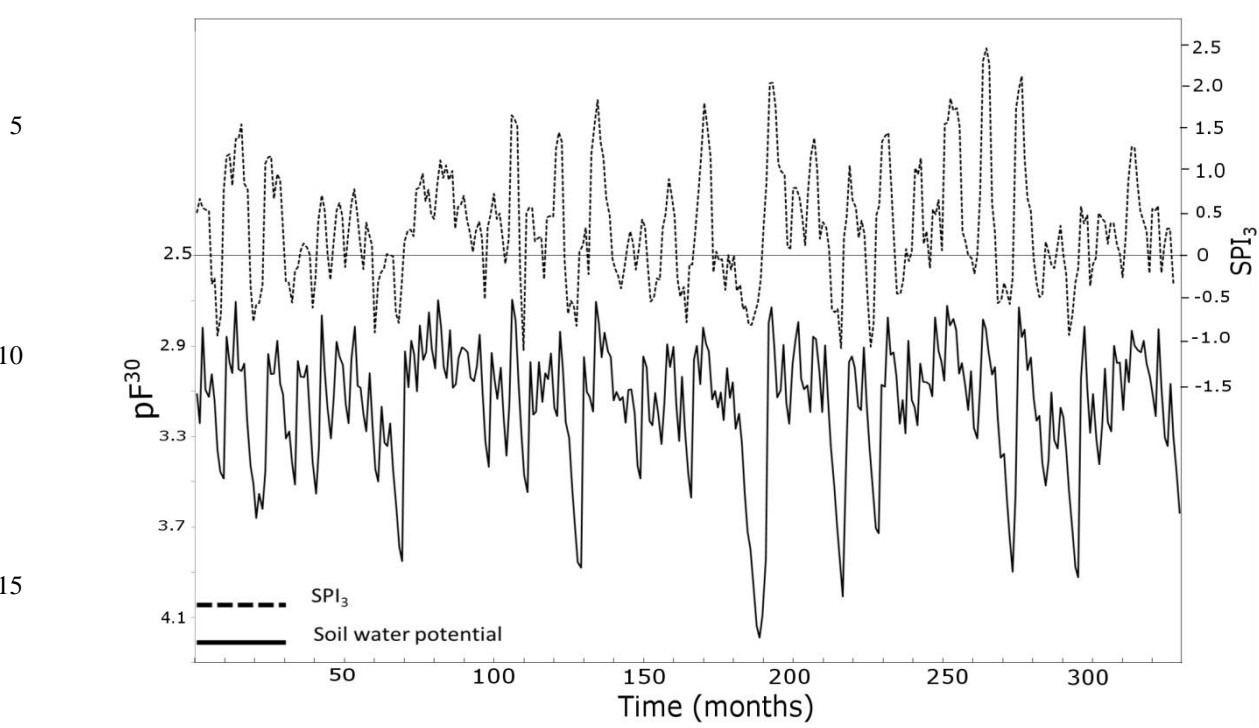

Figure 4: Simulated monthly minimum soil water pressure over 30 cm depth and SPI for Bourke.





Figure 5: Correlations between simulated monthly minimum soil water pressure pF Vs three month time scale of SPI and RDI for 5 cm and 30 cm soil depth. The scatter plots represent the highest correlation for each location.





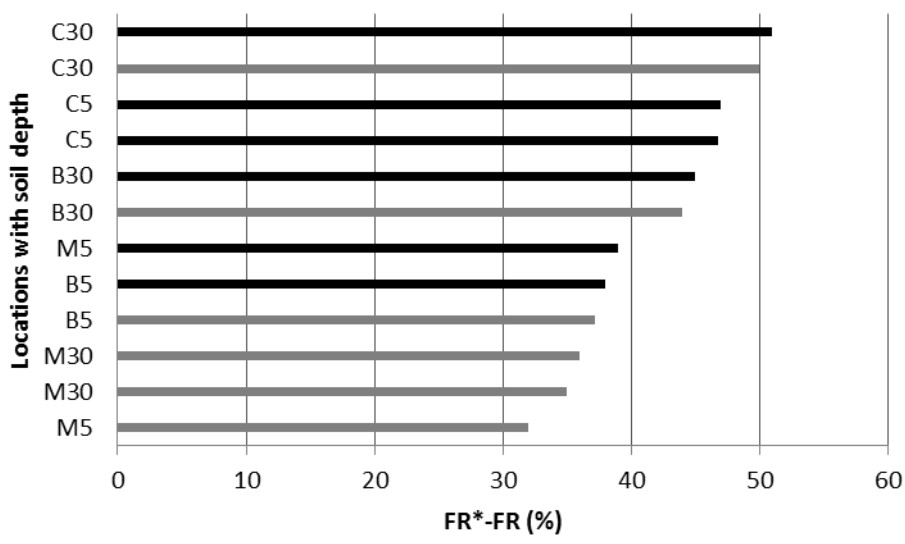

Figure 6: The difference between the FR*-FR for all sites and soil profiles for SPI. The positive values indicate that drought indices are more reliable than the hydraulic model due to the uncertainties of the model. Black and grey bars indicate the highest sensitive and median sensitive values of perturbed model respectively. Values of FAR*-FAR are the same as FR*-FR. FR* represent the mean of three assessing parameters ($\theta r$, $\theta s$, $\alpha$).





**Appendices**

**Appendix A. Configuration of the Hydrus-1D model for all selected sites**

Table A: Configuration of the Hydrus-1D model for all selected sites.

| Attribute | Value |
|---|---|
| *Soil Profile* | |
| Depth (mm) | 50 mm or 300 mm |
| No. of layers | 1 |
| No. of nodes | 100 |
| Nodal density | 100 (upper), 1 (lower) |
| *Hydraulic model and boundary conditions* | |
| Single Porosity model | Van Genuchten-Mualem |
| Hysteresis | No hysteresis was included |
| Upper boundary | Atmospheric (rainfall and evaporation data) with Surface runoff |
| Lower boundary | Free drainage |
| *Iteration criteria and time information* | |
| Maximum No. of iterations | 10 |
| Water content tolerance | $10^{-5}$ |
| Pressure head tolerance (mm) | 1 |
| Lower [upper] optimal iteration range | 3 [7] |
| Lower [upper] time step multiplication factor | 1.3 [.7] |
| Lower [upper] limit of the tension interval (mm) | $10^{-5}$ [$10^6$] |
| Initial [final] time (day) | 0 [9125] |
| Initial time step (day) | $10^{-3}$ |
| Minimum [maximum] time step (day) | $10^{-5}$ [1] |





**Appendix B. Schematic of periods of the simulated soil water pressures in relation to the calculated drought index and schematic of periods of the perturbed Hydrus parameters in relation to the default Hydrus parameters.**

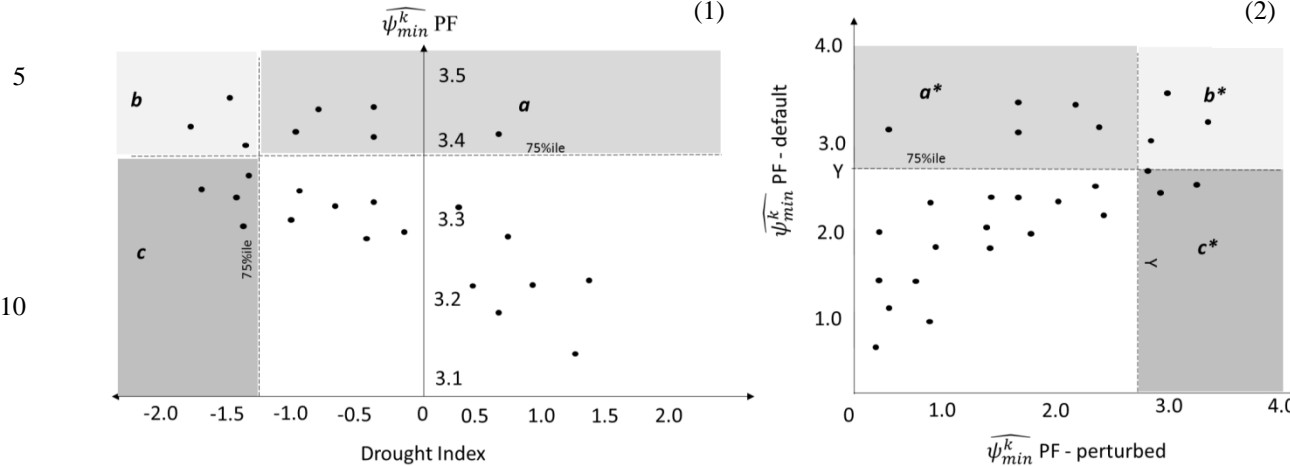

Figure B: (1) Schematic of periods of the simulated soil water pressures in relation to the calculated drought index. The threshold values (dashed lines) divide the schematic into four segments, upon which the failure rate (FR) and false alarm rate (FAR) are based on (Section 2, step 3). The segments represent the simulated low soil water pressure events that are not (a) or are (b) detected by the drought index, and the drought events detected by the drought index that do not (c) or do (not labelled) correspond to periods of low simulated soil water pressure. (2): Schematic of periods of the perturbed Hydrus parameters in relation to the default Hydrus parameters. The threshold values (dashed lines) for default is 75[th] percentile and the threshold value for perturbed is the soil water pressure value of 75[th] percentile of default (y).



**Appendix C. Normalised relative sensitivity S of simulated soil water pressure to uncertainty in the water retention curves.**

Table C: Normalised relative sensitivity S (Eq. 5) of simulated soil water pressure to uncertainty in the water retention curves.

| Site and the soil depth | Assessing parameter | | Percentage of change in the parameter | | | | | | | | | |
|---|---|---|---|---|---|---|---|---|---|---|---|---|
| | | | 50%- | 40%- | 30%- | 20%- | 10%- | 10%+ | 20%+ | 30%+ | 40%+ | 50%+ |
| **Bourke 5 cm** | θr | 0.07 | 0.38 | **0.97** | 0.57 | 0.58 | 0.84 | *0.44* | 0.15 | 0.06 | 0.02 | 0.01 |
| | θs | 0.33 | 0.36 | **1.02** | 0.55 | 0.55 | 0.77 | *0.36* | 0.11 | 0.03 | 0.00 | 0.02 |
| | α | 0.02 | 0.36 | **0.95** | 0.54 | 0.53 | 0.74 | *0.33* | 0.09 | 0.02 | 0.01 | 0.03 |
| | n | 1.30 | NA | NA | NA | NA | **2.45** | 2.34 | *0.94* | 0.54 | 0.36 | 0.26 |
| **Bourke 30 cm** | θr | 0.07 | 0.50 | 0.66 | 0.70 | 0.90 | **1.57** | 1.24 | *0.56* | 0.33 | 0.23 | 0.16 |
| | θs | 0.33 | 0.49 | 0.67 | 0.69 | 0.87 | **1.51** | 1.18 | *0.52* | 0.31 | 0.21 | 0.15 |
| | α | 0.02 | 0.50 | 0.67 | 0.68 | 0.85 | **1.47** | 1.13 | *0.50* | 0.29 | 0.19 | 0.14 |
| | n | 1.30 | NA | NA | NA | NA | **2.69** | 2.36 | *1.06* | 0.65 | 0.45 | 0.34 |
| **Cairns 5 cm** | θr | 0.05 | 1.14 | 0.63 | 2.50 | 3.51 | *2.26* | **3.75** | 2.85 | 0.65 | 0.23 | 1.37 |
| | θs | 0.47 | 1.10 | 0.56 | 2.50 | 3.51 | *2.18* | **4.02** | 2.82 | 0.62 | 0.20 | 1.36 |
| | α | 0.19 | 1.12 | 0.61 | 2.46 | 3.45 | *2.18* | **3.61** | 2.79 | 0.61 | 0.18 | 1.35 |
| | n | 1.21 | NA | NA | NA | NA | NA | **2.86** | 3.22 | *2.03* | 0.73 | 0.40 |
| **Cairns 30 cm** | θr | 0.05 | NA | NA | NA | 2.77 | **3.33** | 2.24 | *1.42* | 0.73 | 0.42 | NA |
| | θs | 0.47 | NA | NA | NA | 2.39 | **2.85** | *2.16* | 1.47 | 0.72 | 0.40 | NA |
| | α | 0.19 | 0.24 | 0.35 | 0.81 | 2.60 | **3.20** | 2.14 | *1.37* | NA | NA | NA |
| | n | 1.21 | NA | NA | NA | NA | NA | **1.44** | 0.51 | 1.06 | *0.74* | 0.46 |
| **Melbourne 5 cm** | θr | 0.10 | 0.08 | 0.03 | 0.16 | 0.37 | **0.96** | 0.11 | *0.25* | 0.27 | 0.26 | 0.24 |
| | θs | 0.51 | 0.06 | 0.05 | 0.19 | 0.42 | **1.07** | 0.34 | 0.21 | *0.26* | 0.26 | 0.25 |
| | α | 0.11 | 0.06 | 0.07 | 0.21 | 0.46 | **1.14** | 0.06 | 0.34 | 0.32 | *0.30* | 0.27 |
| | n | 1.37 | NA | NA | NA | 0.33 | **0.30** | *0.16* | 0.16 | 0.16 | 0.22 | 0.07 |
| **Melbourne 30 cm** | θr | 0.10 | 0.08 | 0.13 | 0.19 | 0.32 | 0.70 | **0.74** | 0.38 | *0.28* | 0.22 | 0.19 |
| | θs | 0.51 | 0.08 | 0.13 | 0.20 | 0.34 | 0.74 | **0.74** | 0.40 | *0.29* | 0.23 | 0.20 |
| | α | 0.11 | 0.08 | 0.14 | 0.22 | 0.36 | 0.80 | **0.86** | 0.45 | *0.32* | 0.26 | 0.22 |
| | n | 1.37 | NA | NA | NA | NA | 0.35 | **0.45** | *0.25* | 0.18 | 0.14 | 0.10 |

Note: Target parameter: daily mean soil water potential, N/A– model cannot run implying implausible value. Values in bold and italics are the highest and the median values of percentage change over all 10 perturbations