# Peer review of "Reliability of meteorological drought indices for predicting soil moisture droughts"

_Hydrology and Earth System Sciences, 2016_

## Referee Comment (RC1) · Anonymous Referee #1 · 7 Oct 2016

General comment The paper tests the capability of simple meteorological drought indices to detect drought events, as defined by simulated soil moisture time-series. The topic is of interest for practical applications in drought monitoring, since simulations are often hard to be performed over some areas.

My opinion is that the overall quality of the paper is negatively affected by some basic assumption made by the authors during the analysis, which are not clearly presented and sometime poorly described. Often the reported results seems off, due to errors or unclear explanations. Hence, I suggest to the authors to carefully reread the paper before to proceed with a full evaluation of the paper.

Specific comments

First of all, they compared the 3-month SPI and RDI against a time-series of monthly

minimum pF. If I have understood correctly, this means that this time series is obtained by choosing the minimum pF value (out of roughly 30 values) for each month in the simulation period. If this is the case, I'm really surprised to see the really good correspondence between SPI and pF as shown in Fig. 4 (and 5 as well). Since SPI (as well as RDI) is a standardized variables, its "random" behavior in Fig. 4 is justified, but the same cannot be said for minimum pF which should retain a sort of seasonality depending on the climate of the area. I'm not familiar with the climate of the specific study region, so it is possible that this behavior is due to the peculiar climate of the region, but in general it is not advisable to perform a correlation analysis between a standardized variable (SPI) and a non-standardized one. This should lead to unfocussed readers to assume that this approach is valid also over other regions.

Also, the authors do not clarify if the 75% threshold is computed separately for each month or for the whole dataset. I assume is the first case (based on the data in Fig. B), but this is never clearly stated. Following this topic, in the same figure it seems that the 75% threshold corresponds to an SPI value around 1.2. This means either that: 1) both tails of the distribution are accounted in this computation, but the correct approach would be to consider just one tail since drought event (i.e., extreme dry conditions) are analyzed here, or 2) the fitting of you distribution is poor since the theory suggests that only about 11% of the data should be $< -1.2$ according to the normal distribution (about 3 values). It is fundamental that this issue is clarified and eventually fixed.

The analysis on extreme values is really misleading, and it also needs to be extended by including other indices. The authors say that FR and FAR are identical in all the cases, but this shouldn't be the case. FR is equal to FAR only if a and c are the same, but this is really unlikely to happen in real cases. For instance, in your example in Fig. B (which I assume is from one of your cases): FR = 5/8=62.5% whereas FAR = 4/7 = 57.1%. Please recheck your calculation of those indices. Also, FR and FAR are not the only indices relevant in this case, e.g., what about the skill of the SPI? Is it better than the climatology or the random case?

Finally, the results of the sensitivity analysis are surprising and need some clarifications. In almost all the case you have FR/FAR values between 30 and 50% higher than in the case of SPI. This means that FR/FAR values for the perturbed simulation are in the order of 65-70% in all cases, included several cases where only a 10% error in 1 parameter is added/subtracted (ie., Bourke 5 cm, Cairns 5 cm, Melbourne 5 cm). I'm really surprised by this result, since in my experience, even for a very sensitive parameter, a 10% change can rarely leads to have 2/3 of the previously detected extremes not detected anymore. It would be useful to have a figure with the reference and perturbed simulations (only the maximum and median ones), as well as the corresponding threshold values, in order to better understand how these changes affect the results. Also, judging from Fig. B it seems that the same 75% threshold is used for both the reference and the perturbed simulation. I assumes that this is not the case, and it is just a coincidence, but I suggest to clarify in the text that the 75% threshold is adapted for each simulation accordingly to the simulated values.

Minor comments

P1, L6. Replace evapotranspiration with potential evapotranspiration.

P1, L6. Rephrase as "used as proxy of severity and duration…."

P1, L15. "…the frequency with which the simulated… below threshold". Actually, you do not want to estimate the frequency, since the frequency is already known as soon as the threshold is defined. Please rephrase.

P2, L7-8. "…water is controlling…. (e.g., water cycle)". Please rephrase.

P3, L6. I would rephrase as something like "The analysis …." Since you are not actually strictly describing a "method".

P3, L 27. How many years were used for the fitting? Which period (the full period?). Please clarify. Also, you should say something about the quality of the fittings (the same is true for RDI).

P4, L21. Appendix A is just a table. Do you really need an appendix for a table? Same for the other appendices.

P4, L22. Remove the parenthesis before "Australian" and move it before "2011".

P5, L1. Please clarify if minimum means minimum among the 30sh daily values in a specific month. Also, please include a standardization of the variable for the successive comparisons with SPI, RDI.

P5, L8-16. Please add at least a skill score.

P5, L17-24. This part on the definition of the threshold is unclear. Please clarify if the threshold is calibrated or not, since you contradict yourself successively in the text. Also, is the threshold computed for each month separately (e.g., 12 thresholds) or for the whole year? The first would be definitely better for the pF.

P5, L20. In Arnold et al. (2014) is reported that there is still seeding also at the wilting point, which does not means that there is no stress. The capability to germinate is clearly reduced compared to optimal water conditions. You should check Cammalleri et al. (2016) "A novel soil moisture based drought severity index (DSI) combining water deficit magnitude and frequency" where a combination of water stress and frequency is used to define drought from simulated soil moisture. Your definition based only on frequency can lead to erroneous estimates over wet areas.

P5, L21. "all values below zero...". It seems that this is not what was done since the 75% threshold is not at zero.

P5, L22-24. This statement is true only in theory (see Fig. B) and it also highlights how it does not make sense to test two indices if is known a-priori that they would be the same.

P6, L9. How do you define the "most extreme droughts"?

P6, L10. How was the interval -0.5, 0.5 chosen? Also, please report that 10% steps

were adopted.

P6, L9-14. To compare extreme pF values in different sites does not make much sense, since one site can be "naturally" drier than another which is not related to the occurrence of a drought event (e.g., pF in a dry area after a rainy period can be higher that pF during a drought in Sweden). This is the reason why standardized SPI is used.

P6, L28-29. This contradict what stated in the methodology.

P8, L28-30. The FR alone cannot fully explain the performance of SPI. For instance, is this better that randomly guessing drought events? E.g, How skillful is this index?

P9, L20. Is more representative compared to what? (I assume to PDSI considering the reference). I would rephrase as "it represent well. . .".

P10, L9. "rather well. . . then. . .". Please rephrase.

P10, 18-19. This sentence is not clear; also, Fig. 5 seems not relevant to this discussion.

P11, L11-12. This is not necessarily the case. If a significant trend in soil moisture is observed on the site, the use of a longer time-series could negatively affect the analysis (without proper de-trending, etc.).

Fig. 4. Please report the starting/ending dates, as well as a time scale that is multiple of 1 year (e.g., 12-24, 48. . .) to make the figure more readable.

Fig. 5. "The plots represent the highest correlation". What does it means? Please clarify.

Fig. 6. Please re-arrange this figure to make it clearer. E.g., order for site and depth, etc.. Also, the acronym C30, M5, etc. are not defined. Please clarify.

---

## Author Comment (AC1) · 21 Oct 2016

We would like to thank the Anonymous Referee #1 for this review and the constructive comments. We would like to take the opportunity to address the four major concerns immediately and address the minor comments in a more detailed response later.

General comment

The paper tests the capability of simple meteorological drought indices to detect drought events, as defined by simulated soil moisture time-series. The topic is of interest for practical applications in drought monitoring, since simulations are often hard to be performed over some areas. My opinion is that the overall quality of the paper is negatively affected by some basic assumption made by the authors during the analysis, which are not clearly presented and sometime poorly described. Often the

reported results seem off, due to errors or unclear explanations. Hence, I suggest to the authors to carefully reread the paper before to proceed with a full evaluation of the paper.

Response: All the authors are confident that the paper is ready for review. We regret and partly disagree fundamentally with some of the referee's views. We hope we can clarify the referee's major misunderstandings – see our responses below - and we await with interest the comments of other reviewers.

Specific comments

Comment 1: First of all, they compared the 3-month SPI and RDI against a time-series of monthly minimum pF. If I have understood correctly, this means that this time series is obtained by choosing the minimum pF value (out of roughly 30 values) for each month in the simulation period. If this is the case, I'm really surprised to see the really good correspondence between SPI and pF as shown in Fig. 4 (and 5 as well). Since SPI (as well as RDI) is a standardized variables, its "random" behavior in Fig. 4 is justified, but the same cannot be said for minimum pF which should retain a sort of seasonality depending on the climate of the area. I'm not familiar with the climate of the specific study region, so it is possible that this behavior is due to the peculiar climate of the region, but in general it is advisable to perform a correlation analysis between a standardized variable (SPI) and a non-standardized one.

Response 1: SPI and RDI are not standardised seasonally in this application (Eqs. 1 and 2) and therefore they do include seasonal patterns. In this regard the good correspondence between the SPI and pF should be no surprise. In Figure 4 we plotted the simulated soil water pressure for Bourke, where the ratio between winter and summer rainfall is 0.61 (Table 1). The seasonality patterns are much more distinct Cairns though (see revised Fig. 4 - included below), where the ratio is 0.10. We strongly disagree that "in general it is not advisable to perform a correlation analysis between a standardized variable (SPI) and a non-standardized one". The definition of any index is

rather arbitrary and usually specific to the pre-defined problem. That said, we believe it is essential to compare indices with physically measurable and plausible variables – no matter if the index includes a standardisation process or not. Ideally, indices such as the SPI or RDI are compared with empirical field data. However, for a variety of reasons such long-term monitoring programs are restricted to limited funding and time. Therefore, a logical step before implementing any long-term campaigns is to test their feasibility in a desktop study using physically based models such as Hydrus-1D with available empirical data such as rainfall/evaporation and soil water retention characteristics. We believe our study covers exactly this critical step!

Comment 2: Also, the authors do not clarify if the 75% threshold is computed separately for each month or for the whole dataset. I assume is the first case (based on the data in Fig. B), but this is never clearly stated. Following this topic, in the same figure it seems that the 75% threshold corresponds to an SPI value around 1.2. This means either that: 1) both tails of the distribution are accounted in this computation, but the correct approach would be to consider just one tail since drought event (i.e., extreme dry conditions) are analyzed here, or 2) the fitting of you distribution is poor since the theory suggests that only about 11% of the data should be < -1.2 according to the normal distribution (about 3 values). It is fundamental that this issue is clarified and eventually fixed.

Response 2: The assumption that the 75% threshold is computed separately for each month is wrong. In Section 2.3 (P5 L 17-18) we state that "For each site the threshold that determines a soil moisture drought event was selected by the percentile of all simulated pF5 and pF30 . . ." Further the referee has assumed that the 75% threshold is computed separately for each month based on Figure B. We regret that the conceptual schematic in Fig. B was considered as empirical data and will clarify that in a revised version of the manuscript. In this regard, the further comments of the referee under comment 3 are irrelevant.

Comment 3.1: The analysis on extreme values is really misleading, and it also needs to

be extended by including other indices. The authors say that FR and FAR are identical in all the cases, but this shouldn't be the case. FR is equal to FAR only if a and c are the same, but this is really unlikely to happen in real cases. For instance, in your example in Fig. B (which I assume is from one of your cases): FR = 5/8=62.5% whereas FAR = 4/7 = 57.1%. Please recheck your calculation of those indices.

Response 3.1: The RDI was excluded from the analysis of extreme values because of the better performance of the SPI (Table 3). The RDI results may easily be added to the Appendices though upon further reviewers' comments. We regret that the reviewer has been misled by the unequal number of data points in a+b and b+c, which will be addressed in a revised manuscript.

Comment 3.2: Also, FR and FAR are not the only indices relevant in this case, e.g., what about the skill of the SPI? Is it better than the climatology or the random case?

Response 3.2: It is not clear to us what the reviewer suggests by the 'skill' of the SPI. It will be easy to add additional performance indices, although our current view is that the visual assessment, R2 values and FR/FAR are sufficient.

Comments 4.1: Finally, the results of the sensitivity analysis are surprising and need some clarifications. In almost all the case you have FR/FAR values between 30 and 50% higher than in the case of SPI. This means that FR/FAR values for the perturbed simulation are in the order of 65-70% in all cases, included several cases where only a 10% error in 1 parameter is added/subtracted (ie., Bourke 5 cm, Cairns 5 cm, Melbourne 5 cm). I'm really surprised by this result, since in my experience, even for a very sensitive parameter, a 10% change can rarely leads to have 2/3 of the previously detected extremes not detected anymore.

Response 4.1: The Richards' equation is used in the Hydrus model (P4 L24). Given the non-linearity of the water retention curve it is not surprising that even ±10% changes in the van Genuchten parameters affect the previously detected values disproportionately (Šimunek et al., 2012).

[Figure]

Comment 4.2: It would be useful to have a figure with the reference and perturbed simulations (only the maximum and median ones), as well as the corresponding threshold values, in order to better understand how these changes affect the results.

Response 4.2: We agree with the referee's suggestion, however we are concerned about the number of graphs (3 sites, 2 depths, 3 model parameters = 18 graphs). Therefore as an example we propose one graph for Appendix D – included below.

Comment 4.3: Also, judging from Fig. B it seems that the same 75% threshold is used for both the reference and the perturbed simulation. I assumes that this is not the case, and it is just a coincidence, but I suggest to clarify in the text that the 75% threshold is adapted for each simulation accordingly to the simulated values.

Response 4.3: The reviewer's assumption is incorrect. The threshold of the perturbed pF is same as the 75th percentile of the default pF, as shown in Figure B and stated in the caption. It would not be a useful performance analysis otherwise. We will emphasise this in the revised manuscript.

Reference Šimunek, J., Van Genuchten, M.T., Šejna, M., 2012. HYDRUS: Model use, calibration, and validation. Transactions of the ASABE 55(4) 1263-1274.

Please also note the supplement to this comment:
http://www.hydrol-earth-syst-sci-discuss.net/hess-2016-467/hess-2016-467-AC1-supplement.pdf
* * *
[Figure]

**Fig. 1.** Figure 4. Simulated monthly minimum soil water pressure over 5 cm depth and SPI for Cairns, Bourke and Melbourne.

[Figure]

**Fig. 2.** Figure D. Default and perturbed (median and maximums for parameter alpha) monthly minimum soil water pressure over 5 cm depth for Bourke.

---

## Referee Comment (RC2) · Anonymous Referee #2 · 26 Oct 2016

The present study analyses the reliability and effectiveness of SPI/RDI in predicting soil moisture droughts. The research has significance in agriculture drought monitoring in places without adequate soil moisture observations. However, some revision comments are proposed below:

1. As the key idea about the research is testing the ability of meteorological drought indices in predicting soil drought. Why only use soil water pressure to quantify the 'soil moisture droughts'? I suggest the author should use the observed soil moisture to test the capability of these drought indices. You may not use the SM data of all layers studied, at least the average condition of SM and its correlation with the drought indices should be revealed.

2. As agricultural drought or ecodrought are usually measured by soil moisture. The

relationship between soil moisture and soil water pressure used in current research should be further studied in the 3 stations.

3. I suggest the author also analyze the effect of drought timescale on soil moisture. You may analyze more on soil moisture and drought with changing timescales e.g., 1-12months.

4. In addition to model parameter setting, the input of the model including the climatic data should be clarified to enhance the comparison between model output and the drought indices calculated from precipitaton/PET.

5. In discussion, the author mentioned that 'our results point to the simplest being the best'. Such kind of expression should be very careful as the study only analyses SPI and RDI. Actually there are many effective drought indices with precipitation and PET, e.g. SPEI. The author can read more literatures on this.

---

## Short Comment (SC1) · 30 Oct 2016

Manuscript hess-2016-467 by Halwatura et al.: Reliability of meteorological drought indices for predicting soil moisture droughts

Reviewed by Danny Heuvelink

"Note to the editor and authors: As part of an introductory course to the Master programme Earth & Environment at Wageningen University, students get the assignment to review a scientific paper. Since several years, students have been reviewing papers that are in open online discussion for HESS, and they have been asked to submit their reports to the discussion in order to help the review process. While these reports are written as official reviews, they were not requested for by the editor, and we leave it up to the editor and authors to use these reports to their advantage. While several

students were asked to review the same paper, this was not done to provide the authors with much extra work. We hope that these reports will positively contribute to the scientific discussion and to the quality of papers published in HESS. This report was supervised by dr. Ryan Teuling."

This paper evaluates how well the SPI and the RDI perform at estimating soil moisture droughts simulated by a physically based soil water model. The analysis of this paper is based on three sites in Eastern Australia. The soil water pressure was simulated using the Hydrus-1D model. The performance of the two drought indices was measured by calculating the correlation between the indices and the simulated monthly minimum soil water pressures. It was found that there was a significant correlation between the drought indices and the simulated monthly minimum soil water pressure. For most locations and all soil depths the FR and FAR were below 50 %, meaning that the indices are able to capture the occurrence of droughts quite well. SPI performed better in total than RDI in terms of FR and FAR. The uncertainty in the model is quite high, but the model approach produces physically meaningful values which are plant species specific.

The importance of the investigation subject addressed by this paper is quite high. Drought indices are used in a lot of researches, because of their simplicity and their use of easy obtainable measurements like rainfall. It also fits nicely within the scope of HESS, because the importance of understanding and monitoring droughts is very high, especially with changing climate it is more than ever necessary to be able to monitor droughts accurately. This paper also uses techniques and approaches that follow the scope of HESS, like modelling, mathematic applications and uncertainty analysis. Assessing the performance of commonly used drought indices is highly valuable, the interesting part of this paper is that it compares two different drought indices not only to each other, but also to a hydrological based model. This paper can thus potentially add more knowledge on drought indices and the use of a physically based hydrological model, improving the field of Hydrology. The novelty in this paper is especially

the assessment of drought indices performance with the use of a hydrological model, therefore creating opportunities to test the performance of the drought indices without actual soil moisture observations.

However, there are major issues in this paper; the most important issue is the fact that the drought definition of the drought indices and the physically based model is different and therefore their comparison is not as straightforward as treated in this paper, see argument 1. This needs a lot of attention and a drastic change of methods. Also the reasoning behind choosing the 75th percentile as drought threshold is not explained well, the choice for the monthly minimum soil water pressure is not explained and the comparison of the results with other papers is too marginal, see arguments 3, 4 and 5. From the list of minor arguments it becomes clear that a lot of claims or choices made in this research are not explained properly. Also the writing of this paper has to be looked at critically, because of the many small mistakes and incorrect sentences. Based on all this, it becomes clear that this paper cannot be published in its current form.

Argument 1 different drought definitions for indices and the model

The drought indices calculate anomalies from the average condition based on monthly averages over all years in the dataset. This definition makes it that the SPI always addresses drought as a deviation from the average case (Tsakiris and Vangelis, 2005; McKee et al., 1993). For the drought based on the simulation by the Hydrus-1D model, the 75th percentile of the monthly minimal soil water pressure is used as a threshold. This threshold pressure is not based on seasonality but just on the total monthly minimum soil water pressures. So the drought indices are based on differences between the monthly values and the average monthly values, and the simulated minimum monthly soil water pressures are not based on seasonality. Therefore the comparison between these two is not straightforward to make, because they look at different things. The drought indices don't look at absolute droughts, but to relative droughts while the Hydrus-1D model outcomes used in this studies say something about the absolute

droughts.

To improve this part of the research a clear definition of drought has to be given that is used for as well defining drought periods by the drought indices and for the Hydrus 1-D model. Therefore the use of the drought indices or the definition of drought according to the minimum monthly soil moisture pressure has to be altered.

Santos et al. (2013) used the SPI to create maps of a site in Portugal with monthly precipitation that responses with a SPI value of -1.28, the threshold they chose for severe drought. The difference in precipitation between months is very large. Using the same methods as Santos et al., 2013 all drought indices values which are considered to be a drought can be transformed to the rainfall amount it corresponds to. That value is than comparable to the monthly minimum soil water pressure.

My advice is to use the methods Santos el al. (2013) used and transform the SPI and RDI values back into rainfall amounts and compare those with the monthly minimum soil water pressure. For the RDI this would mean to transform it into a net rainfall amount (precipitation minus evapotranspiration). This method would assure that the comparison made between the drought indices and the model would be based on the same definition of drought. This change is very drastically, but necessary, and would in fact change a big part of the methods and all the results and conclusions.

Argument 2 why only SPI and RDI?

There is no clear reasoning why only the SPI and RDI indices are used in the comparison. Why not use more than only these 2 indices. If more indices are used, with all different complexity, a lot more can be said about the performance of drought indices compared to the model. More statements can be made on the success of less complex indices compared to the more complex indices. Other indices that could for instance be included in the research are: standardized precipitation evapotranspiration index (SPEI) and Palmer drought severity index (PDSI). Using additional drought indices would lead to more solid conclusions and better understanding of the performance of

drought indices compared to a physically based model.

Argument 3 why the 75th percentile threshold?

In the paper it is not mentioned why exactly the 75th percentile was chosen to represent the drought threshold. There should be a clear reasoning why a certain threshold was chosen. First of all is there a need to choose a non-physically based threshold. In the methods there is stated that for instance the wilting point does not necessarily coincide with the stress levels of plants. But this is based on one research, Arnold et al. (2014). This paper only researches this for 1 specific plant species. There is no evidence given for other species, so it cannot be said that a physically based threshold is not appropriate. So more research needs to be done on that. When the approach of choosing a percentile is researched and assumed correct, than a clear reasoning has to be given why a certain percentile is chosen. Following are different thresholds used by other papers on which a drought threshold can be chosen.

Agnew (2000) states that the drought threshold for the SPI should be based on the probability of occurrence. Therefore a moderate drought would already be at an SPI of -0.84, or transformed to a percentile the 60th percentile. Mckee et al. (1995) and Komescu (1999) place the threshold for moderate drought at SPI=-1.00 , transformed to percentile this is the 68th percentile.

Svoboda et al. (2002) define different drought classes, namely D0 (abnormally dry, 20–30% percentile), D1 (moderate drought, 10–20%), D2 (severe drought, 5–10%), D3 (extreme drought, 2–5%) and D4 (exceptional drought, <2%). Translated to the percentiles used in this study , it would be the 70th percentile for D0, the 80th for D1, the 90th for D2, the 95th for D3 and the 98th for D4. So depending from which drought severity onwards a drought should be defined, the percentile has to be chosen.

Concluding there are different choices that can be made to define the threshold from which onwards a drought is recognized. Therefore a clear choice of threshold value should be given and it has to be explained. When after proper research it is found
acceptable that a non-physically based threshold is appropriate, the easiest solution would be to keep the 75th percentile as a drought threshold, because then there would be no changes in the conclusions. But add some references like the ones above to highlight that this is an arbitrary choice and many researches use different thresholds.

Argument 4 why the minimum soil water pressure?

Why is the minimum soil water pressure used, no explanation is given on why this is used and this refers to the minimum of the month, but is it not better to take the average monthly soil water pressure? Suppose you are at the end of a drought, but the 1st soil water pressure of a month is still high, but is declining the rest of the month. Than your monthly minimum soil water pressure gives a very high value, even though the average of the month is way lower. The drought index values average over this month, so it is not fair to compare this value with the minimum soil water pressure over that month. Changing the monthly minimum soil water pressure to average soil water pressure causes the results of this research to be changed to some extent, but it is unclear for me how big this effect would be, because I have no insight in the dataset used.

Argument 5 comparison of results

Sims and Raman (2002) are researching the performance of PDSI and SPI in estimating the soil moisture. It turned out that for their study SPI was considered to perform better in estimating the soil moisture than PDSI. But there are a lot of differences between the study of Sims and Raman and this paper. Firstly the comparison in this paper is with RDI and not with PDSI like in Sims and Ramans' paper. Although both include evapotranspiration, PDSI also includes local water availability. Also the study performed by Sims and Raman uses real soil moisture measurements, therefore it is hard to compare the results with this paper, that uses modelled soil water pressure. Also the area for which the researches are performed are different. Sims and Ramans' study is performed for North Carolina and this papers' research for Eastern Australia. So concluding it is very hard to compare the results of the paper of Sims and Ra-

man with this paper and more references should be given to make a proper statement whether different researches came with similar results.

An example of another research that investigated the performance of the RDI and SPI is Shokoohi & Morovati (2015). They compared the performance of the RDI and the SPI for the Lake Urmia basin in Iran. The conclusions of this paper are that in a IWRM framework RDI is favourable over SPI. The paper of Zarch et al. (2015) states that the RDI index is favourable over the SPI index on a global scale in assessing future droughts, because of the increasing temperature trend. Although these researches all use different methods and the assed areas are different, it highlights that the result found for which drought index performs better is strongly influenced by the location and methods used in the research. So more references, like the ones provided above, should be given to compare the results found with others.

Minor arguments

1 The introduction is well structured and sets up the reason to research well.

2 The references to the fact that there are little soil moisture observations need to be more recent, more recent references are for instance Smith et al. (2012) and Peisch et al. (2012).

3 Instead of figure 2 showing only the location of the three selected locations, it can be upgraded by also listing soil type in the figure, to make the reasoning behind the locations chosen more insightful.

4 The layout of the figures and tables is quite plain and the bold and especially italic values are not easily seen. Maybe adding some colour in the tables would make them easier understandable and more appealing. Also the Bold and italic values could be indicated with different colours.

5 In the conclusion the focus lies more on the difference between the SPI and RDI, while in the abstract only 1 sentence is dedicated to this. Therefore adding a few

sentences in the abstract about this difference would make the abstract more representable for the research.

6 In the methods section 2.1, the selected sites are listed, but there is no clear reasoning given why especially these sites are chosen. Please add more explanation on why these sites have been chosen.

7 In the methods section 2.2, the selected average periods are listed and it is said that only the three month period will be shown, for the sake of simplicity. But are there other reasons why to only show the three month period. Please add some more reasoning on why only the three month average is shown.

8 there are several assumptions listed for the soil water modelling, but there is no insight given in whether these assumptions are valid or reasonable. Please add a paragraph in which the validity and reasonability of the assumptions is discussed.

9 the reference to Arnold et al. (2014) is very specific for one seed species and its' response to drought conditions. I advise to also refer to a broader paper about this subject.

10 in methods section 2.4, it is unclear why the simulated soil water pressure is considered sensitive to uncertainty if S>1, adding a few lines that explains this would make this section easier to understand.

11 No explanation is given on why the Hydrus 1-D model is used and not another model, please add some more explanation on why exactly this model is used.

12 I like the fact that an uncertainty analysis on the performance of the Hydrus-1D model has been performed and the discussion of this uncertainty analysis.

13 Figure 1, with the diagram of the steps in the methodology, helps to capture the structure of the methods better, therefore I advise to refer more often to this figure in the methods, to help the reader capture the links between the methods.

14 In the conclusion it is said that a False Rate of 19-32 % is considered to be a satisfactory performance, but in the discussion it is not explained why this is satisfactory, no comparison of these values were made with other studies, so it is hard to say whether these values are good or not.

15 figure 5 does not show the pF 5 values in the correlation graphs, but is said in the caption that it is treated in the figure. This figure is also really hard to understand, because of the missing explanation of what is shown in the figure. Maybe it is wiser to show the R2 values in bar graphs or a table.

Minor issues

p.1, line 20: missing "the" before "drought"

p.2, line 11: the "," after "e.g." has to be removed

p.2, lines 14-16: a reference that confirms this is missing

p.2, line 27: "to" is missing after "questions"

p.3, line 1: "Thorough" has to be changed into "Through"

p.4, line 18: "see below" has to be changed in "see next page"

p.4, line 22: the reference seems out of place here

p.5, line 23: "the" is missing before "same" (twice)

p.5, line 23: "of" has to be changed into "for"

p.5, line 24: "equation" has to be changed into "equations"

p.6, line 15: "uncertainty" has to be changed into "uncertainties"

p.6, line 22: a "," is missing after "(appendix C)"

p.6, lines 24-25: the font size of the equations is not the same

p.7, line 21: this sentence is unclear, maybe "all sited" has to be changed into "for all sites"

p.7, lines 11-14: these lines are should be rewritten, because they are unclear now.

p.7, line 18: a "," is missing after "default values"

p.7, line 19: "equation" has to be changed into "equations"

p.7, line 21: a "," is missing after "42 %"

p.8, lines 1-2: this sentence has to be rephrased to get the point across that with the 75th percentile the FAR and FR are the same, now it this sentence indicates that the FR, or FAR, at each depth and location is equal.

p.8, line 3: "is" is missing after "site"

p.8, line 14: "drought index" has to be replaced with "SPI"

p.8, line 16: "more so" has to be replaced with "especially"

p.8, line 16: "the" is missing after "in"

p.8, line 17-18: "relatively well" should be replaces with "better" and after "than", "for" is missing

p.8, line 28: "at" is missing after "frequency"

p.9, line 4: "for an example" has to be replaced with "for example"

p.9, line 8: "we" is missing after "best"

p.9, line 21: "into to" has to be replaced with "they were used for"

p.9, line 15: "for an example" has to be replaced with "for example"

p.10, line 9: "rather well" has to be replaced with "better" and "for" is missing after "than"

p.10, line 17: "index" has to removed

p.10, lines 17-19: please rephrase this sentence, because it is unclear now

p.10, line 24: "this implies" refers to the previous sentence, but it doesn't match with the information provided in this sentence

p.10, line 29: "in" has to be replaced with ","

p.11, line 17: "in" has to be removed

p.11, lines: 25-26: this sentence is not clear, please rephrase it

p.11, lines: 29-30: this sentence is not clear, please rephrase it

p.11, line 31: "be" is missing before "more"

p.11, line 31: "be" is missing before "more"

p.11, line 32: "Potential advantages" has to be changed into " A potential advantage"

p.16, Table 4: "be" is missing after "may"

p.21, Figure 5: in the caption it is said that the 5 and 30 cm soil depth RDI and SPI are shown, but only the 30 cm RDI and SPI is shown

p.22, line 5: "represent" has to be changed into "represents"

p.25, Table C: not everything is at the right place in the header

References

C. T. Agnew, Using the SPI to Identify Drought, 2000, Vol. 12, No. 1, Winter 1999–Spring 2000, University College London, London, United Kingdom

Arnold, S., Kailichova, Y., andBaumgartl, T., 2014. Germination of Acacia harpophylla (Brigalow) seeds in relation to soil water potential: implications for rehabilitation of a threatened ecosystem. PeerJ 2 e268.

McKee, T.B., Doesken N. J, andKleist John, 1993. The relationship of drought frequency and duration to time scales, Proceedings of the 8th Conference on Applied Climatology. American Meteorological Society Boston, MA: Anaheim, California, pp. 179-183.

McKee, T. B.; N. J. Doesken; and J. Kleist. 1995. "Drought monitoring with multiple time scales." Proceedings of the Ninth Conference on Applied Climatology; pp. 233–236. American Meteorological Society, Boston.

Peischl, S., Walker, J. P., Rüdiger, C., Ye, N., Kerr, Y. H., Kim, E., Bandara, R., and Al-lahmoradi, M.: The AACES field experiments: SMOS calibration and validation across the Murrumbidgee River catchment, Hydrology and Earth System Sciences, Discuss., 9, 2763-2795, doi:10.5194/hessd-9-2763-2012, 2012

J. F. Santos, M. M. Portela, M. Naghettini, J. P. Matos & A. T. Silva, Precipitation thresholds for drought recognition: a further use of the standardized precipitation index, SPI, 2013, WIT Transactions on Ecology and The Environment, Vol 172, doi:10.2495/RBM130011

Alireza Shokoohi & Reza Morovati, Basinwide Comparison of RDI and SPI Within an IWRM Framework Water Resour Manage (2015) 29, 2011–2026, DOI 10.1007/s11269-015-0925-y

Sims, A.P., and Raman, S., 2002. Adopting drought indices for estimating soil moisture: A North Carolina case study.,Geophysical Research Letters 29(8).

Šimůnek, J., van Genuchten, M.T., and Šejna, M., 2008. Development and Applications of the HYDRUS and STANMOD Software Packages and Related Codes. Vadose Zone Journal 7(2) 587-600.

Smith, A. B., J. P.Walker, A. W.Western, R. I.Young, K. M.Ellett, R. C.Pipunic, R. B.Grayson, L.Siriwardena, F. H. S.Chiew, and H.Richter (2012), The Murrumbidgee soil moisture monitoring network data set, Water Resour. Res.,48, W07701, doi:10.1029/2012WR011976.

Svoboda M, Lecomte D, Hayes M, Heim R, Gleason K, Angel J, Rippey B, Tinker R, Palecki M, Stooksbury D, Miskus D, Stephens S (2002) The drought monitor. Bull Am Meteorol Soc 83:1181–1190

Tsakiris, G., and Vangelis, H., 2005. Establishing a drought index incorporating evapotranspiration European water 9/10 3-11.

Mohammad Amin Asadi Zarch, Bellie Sivakumar, Ashish Sharma, Droughts in a warming climate: A global assessment of Standardized precipitation index (SPI) and Reconnaissance drought index (RDI), 2015, Volume 526, Pages 183–195
* * *

---

## Short Comment (SC2) · 31 Oct 2016

Note to the editor and authors: As part of an introductory course to the Master programme Earth & Environment at Wageningen University, students get the assignment to review a scientific paper. Since several years, students have been reviewing papers that are in open online discussion for HESS, and they have been asked to submit their reports to the discussion in order to help the review process. While these reports are written as official reviews, they were not requested for by the editor, and we leave it up to the editor and authors to use these reports to their advantage. While several students were asked to review the same paper, this was not done to provide the authors with much extra work. We hope that these reports will positively contribute to the scientific discussion and to the quality of papers published in HESS. This report was supervised by dr. Ryan Teuling.

The purpose of this research is to compare two meteorological drought indices, the Standard Precipitation Index (SPI) and the Reconnaissance Drought Index (RDI), to a physically based soil water model. The methodology consists of five steps. In the first step, sites are selected, in the second step, SPI and RDI are calculated over one, three and twelve month time periods. Soil water pressures are simulated with the model Hydrus-1D. In the third step, the drought indices are compared to the simulated values. PF values, calculated from simulated minimum soil water pressures for each month, averaged over a depth of 5 and 30 cm are correlated with the SPI and RDI values. In addition, the failure rate (FR) and false alarm rate (FAR) are calculated. In the last two steps, sensitivity of the model is determined and accuracy of the model in case of data uncertainty is assessed. From the results, it seems that SPI and RDI perform quite well, and can be used in case of limited data availability. The relatively high FR and FAR values might be due to simplicity of the indices, time scales and potential errors in the simulations.

As the authors state in the introduction of the manuscript, monitoring and predicting drought is of great importance, especially since the occurrence and severity of droughts is expected to increase due to a future increase in climate variability, affecting agricultural production and ecosystem functioning (IPCC, 2014). In addition, data availability is an issue in many areas. Thus, evaluating simpler drought indices is important and the results of this paper are useful for practical applications, such as regional drought monitoring and water resource management. Much research is done on evaluating drought indices, as is also clear from the manuscript and SPI is often evaluated as meteorological drought index for agricultural or hydrological drought (e.g. Kumar et al., 2016; Sheffield et al., 2014; Sims and Raman, 2002). Furthermore, SPI and RDI are compared by e.g. Khalili et al. (2011).; Banimahd and Khalili (2013) and Shokoohi and Morovati (2015). In my opinion, it is interesting that in this research not only SPI are RDI are evaluated (research question 1), but the authors also aim to assess the effect of data uncertainty on simulations (research question 2). In this way, the research provides important information for regions with limited data-availability. The research

corresponds to the scope of the journal, as it addresses drought, the temporal, and to some extent spatial, characteristics of precipitation and soil water pressures, and serves water management. The structure of the manuscript is good and the writing is clear. Some minor comments on the structure are provided in the last section of this review. In my opinion, there are some major issues in the methodology that should be addressed or further explained by the authors before the manuscript is published. I will elaborate on this in the following paragraphs.

Specific comments

Comment 1: In the third step of the methodology, SPI and RDI values are correlated with pF-values. SPI is defined by McKee et al. (1993) as: "Standardized precipitation is simply the difference of precipitation from the mean for a specified time period divided by the standard deviation where the mean and standard deviation are determined from past records." A probability function is defined for a specific time period (over periods of e.g. one month, three months, twelve month) and e.g. a three-month accumulated precipitation is compared to average precipitation over that same period in a range of years. This is supported by e.g. Lyon et al. (2011), Kumar et al. (2016) and Guttman (1999). As a result, SPI indicates anomalies from a certain seasonal precipitation pattern in a specific climate, that occurs in a certain region. The pF values that are calculated from the minimum water pressures for each month and site and averaged over a depth of 5 and 30 cm, do include seasonal variation and climatic differences. Therefore, it is strange that SPI is directly correlated to these pF values. The difference in SPI and pF values will probably have an effect on the correlation. This expectation is supported by the fact that according to the climate index Rw/RsÂň in Table 1 (P15), there is a seasonal pattern in rainfall. I recommend to standardize the pF values, see e.g. Kumar et al. (2016), who make a standardized index for groundwater measurements or Sheffield et al. (2014) (see P4), who create a drought index based on a cumulative probability of soil moisture fitted to simulated values. To ease comparison, the data might be normalized (to a range of [0,1], e.g. Sims and Raman, 2002). If the authors

want to correlate SPI and RDI directly to the pF values, as pF is a direct indication of plant stress, I think it would be good to state in the methodology why they want to do this, what the effect on correlation is and to give an indication how large the seasonality is, e.g. by a graph. Figure 4 (P20) shows pF values and SPI values for Bourke, but the time period is so big, that yearly seasonality is hard to subtract from this graph. In addition, if pF is not standardized, the 75th is not appropriate. In my opinion, it makes sense that for the SPI a certain threshold is defined (P5, lines 17-24), and that this is equal for all sites, as SPI describes anomalies not absolute values. Values of SPI usually get assigned a certain category of drought intensity (McKee et al., 1993). In the methodology, the percentile for pF values is also set at 75. pF values are physically based and are directly related to available soil moisture, plant stress etcetera. In a wet climate, pF values are generally lower than in a dry climate. So the 75th percentile represents a different range of pF values for the different sites and could in a wet climate include pF values that do not lead to plant stress, whereas in a dry climate, exclude values that do lead to plant stress. I recommend that, if the authors compare SPI and RDI to the unstandardized pF values, that a threshold is defined that is set at a certain pF value and is equal for the three sites. In that case, the values of FR and FAR are no longer equal.

Comment 2: When reading the manuscript, for several sections I found myself wondering why certain decision where made or whether there had been previous research on certain topics. I will address four sections of the manuscript where, in my opinion, further explanation and references are required.

First of all, I expect that SPI, and to a lesser extent RDI, have been compared to more complex models and drought indices before. In the discussion, Sims and Raman (2002), Khalili et al. (2011) and other relevant studies are referenced , but in the introduction, no prior research on this is named. It would be good to embed this research in previous research while stressing the novelty of this research. This might also strengthen the methodology, in case other researchers use similar approaches. In

the second paragraph of the review, I named some prior research. However, there are many more studies.

Secondly, from the introduction and methodology it is unclear to me whether the model Hydrus-1D or a comparable model is commonly used for drought monitoring and prediction. If this is the case, it would be good to emphasize this, as it stresses the relevance of comparing SPI and RDI with this particular model and assessing the effect of data uncertainty on the model output. There are studies on effects of drought, e.g. Hartman et al. (2012) on the effect of a.o. climate change on the soil water balance in relation to tillage, and Rahman et al. (2015) on the effect of drought on salt accumulation, but I could not find research on the application of Hydrus for drought monitoring.

Thirdly, it is unclear to me if Hydrus-1D is accurate enough to simulate soil water pressures and thus a proper replacement of soil moisture measurements. The authors state that there might be errors in the simulated soil water data (P9, line 10; P10, lines 12-17), while on the other hand, they assume the simulated water pressures accurate enough for the further analysis (P4, lines 28-29). The best solution would be to validate the model for the sites with actual measurements. This has been done priory, for example by Sheffield et al., (2014), who compare simulated soil moisture values with field measurements, before the values are transformed to a drought index. Another option, would be to at least provide some clarity on the accuracy of the model and how large the effect of errors on the correlation with SPI and RDI is.

Finally, in section 2, methods (P3-7), some argumentation and references on specific steps of the methods would be beneficial. A reference could be given for the FR and FAR in step 3 (e.g. Wilks et al., 2011, p.264), argumentation and references for the method to assess the model sensitivity could be given (step 4, but also affecting step 5), especially since the authors give some more suggestions for uncertainty analysis in the discussion (P10, lines 19-21).

Comment 3: In the fifth step of the methodology, water pressures are simulated for

perturbed parameter values. FR* and FAR* values are calculated and compared to FR and FAR. My issue with this step in the methodology, is that it is unclear to me if the perturbation of the parameters corresponds to data uncertainty that can be expected in reality. Therefore, I cannot be sure if the statement "If FR*>FR or FAR*>FAR, the assumed parameter uncertainty in the hydraulic model critically affects its relative ability to detect droughts, so that the simple drought index may be preferred over the more complex soil water model, even if FR or FAR is high." (P7, lines 3-5) is true. The results of this comparison are shown in Figure 6 (P22), and the difference between FR* and FR is high, but is hard to interpret as it is not defined how realistic the data uncertainty is and in what circumstances this will occur. It is obvious that if data is absent in an area and cannot be obtained, the simpler drought indices are preferred. But if data is available, but uncertain, when is the uncertainty so high that simulation modelling should not be commenced? I recommend that the authors give an indication of what data uncertainty can be expected and either give an indication how close the perturbed parameters in Appendix C are to reality or redo the calculations for more realistic data uncertainty.

Minor comments

P3, Step 1 of the methods: To me it would seem more logical to describe the selected sites, than to state that selection is part the methodology. It the selection is part of the methodology, I would expect more information on e.g. selection criteria. I recommend to describe the sites at the beginning of the methods instead of naming it step 1 in the methodology.

P3, Equation 1: It seems as if the non-exceedance probability is multiplied with the three-month average value, which is probably not the case (P4, line1: "FR is non-exceedance probability of the three-month average value"). If so, the equation should be adapted.

P5, lines 20-21: "At the 75th percentile all drought index values below zero were taken

as drought events." This sentence is unclear to me.

P9, lines 18-20: "However, notwithstanding the limited scope of this paper, our results point to the simplest being the best. Similar results have been observed in North Carolina showing that SPI is more representative of soil moisture variation (Sims and Raman, 2002)". Sims and Raman (2002) compare only two indices, SPI and PDSI and compare for a specific goal: short term variation in soil moisture. The research does not provide enough support for the statement that the simplest is the best. I suggest that the sentence is rephrased, or that additional references are given to support the statement.

P10, lines 7-9: "Given the more rapid variations in soil water content experienced in the shallow soil, if drought indices are used as a planning tool for initial establishment of seeds, a shorter time scale than three months is likely to be preferred." If so, then it might be beneficial to show the results for the 1 month time period calculations.

P10, lines 22-24: "Overall, SPI performed better than RDI, illustrating that in general the inclusion of PET in RDI confounds the prediction of drought for these sites, although this result may be affected by PET estimation and Hydrus model errors as discussed above." The discussion on comparing SPI and RDI could be embedded in prior research, see e.g. Khalili et al. (2011) (referenced in the manuscript), Banimahd and Khalili (2013) and Shokoohi and Morovati (2015).

P11, lines 22-23: "The study reveals that a simple drought index, SPI, which uses only monthly precipitation as an input, may out-perform a more sophisticated index, RDI, over a range of soil types and climates." To me, it was not instantly clear that SPI and RDI would be compared to each other as opposed to the physically based model (P2, lines 26-30). In addition, the difference between performance of SPI and RDI is discussed by the authors: "... although this result may be affected by PET estimation and Hydrus model errors as discussed above." (P10, line 23) and "For arid Bourke, there was less benefit using SPI rather than RDI (Fig. 5), which we speculate is due

to a stronger and more linear influence of PET on the soil moisture minima at that site (Khalili et al., 2011; Asadi Zarch et al., 2015)." (P10, lines 25-27) and is contradicted by previous research (e.g. Khalili et al., 2011). This sentence might be rephrased to: "In this study the simple drought index, SPI, which uses only monthly precipitation as an input, out-performed the more sophisticated index, RDI".

P18, Figure 2: To have some attention for the very small details, the rectangle in the smaller scaled picture (showing Australia) does not exactly correspond to the large scale map and the North arrow in the small figure (showing Australia) might be excluded, as it is probably clear that the orientation of both maps is equal.

P20, Figure 4: This figure is referenced in the manuscript in the following sentence "Drought index values were greatest (most negative) in arid Bourke (-2.94), similarly the pF values for both soil profiles were also highest in Bourke (pF5 = 4.47 and pF30 = 3.38) compared to other two sites (Fig. 4)." (P 7, lines 11-12) However, this is not clear from the figure, as only results for Bourke are shown and it is hard to assess the exact value from the figure. Therefore, I think this figure does not serve its purpose in regard to this sentence. The authors might add a sentence that states what is in the figure (e.g. long term pF30 and SPI values for Bourke).

P21, Figure 5: This figure is to some extent unclear to me. It shows the correlation for pF values and SPI, supplemented with R2 values. This is clear, and the distinction between SPI and RDI using colours is good. However, the graph in the middle is not well explained. For every site, two values are shown, which probably represent correlation with pF30 and pF5, but this is not explained.

Some grammatical errors: e.g. P1, line 21 "the model provide"; P5, line 20 "We choose the 75th percentile to represents"; P7, line 9 "... showed reliable all sited". There are more grammatical errors, but as I understood, the manuscript will be proofread before publishing.

References

Banimahd, S. A., & Khalili, D. (2013). Factors influencing Markov chains predictability characteristics, utilizing SPI, RDI, EDI and SPEI drought indices in different climatic zones. Water resources management, 27(11), 3911-3928.

Guttman, N. B. (1999). Accepting the standardized precipitation index: A calculation algorithm1. Hartmann, P., Zink, A., Fleige, H., & Horn, R. (2012). Effect of compaction, tillage and climate change on soil water balance of Arable Luvisols in Northwest Germany. Soil and Tillage Research, 124, 211-218.

IPCC (2014). Summary for policymakers. In: Climate Change 2014: Impacts, Adaptation, and Vulnerability. Part A: Global and Sectoral Aspects. Contribution of Working Group II to the Fifth Assessment Report of the Intergovernmental Panel on Climate Change [Field, C.B., V.R. Barros, D.J. Dokken, K.J. Mach, M.D. Mastrandrea, T.E. Bilir, M. Chatterjee, K.L. Ebi, Y.O. Estrada, R.C. Genova, B. Girma, E.S. Kissel, A.N. Levy, S. MacCracken, P.R. Mastrandrea, and L.L.White (eds.)]. Cambridge University Press, Cambridge, United Kingdom and New York, NY, USA, pp. 1-32.

Khalili, D., Farnoud, T., Jamshidi, H., Kamgar-Haghighi, A. A., & Zand-Parsa, S. (2011). Comparability analyses of the SPI and RDI meteorological drought indices in different climatic zones. Water resources management, 25(6), 1737-1757.

Kumar, R., Musuuza, J.L., Van Loon, A.F., Teuling, A.J., Barthel, R., Ten Broek, J., Mai, J., Samaniego, L., Attinger, S. (2016). Multiscale evaluation of the Standardized Precipitation Index as a groundwater drought indicator. Hydrology and Earth System Sciences, 20 (3), pp. 1117-1131

Lyon, B., Bell, M. A., Tippett, M. K., Kumar, A., Hoerling, M. P., Quan, X. W., & Wang, H. (2012). Baseline probabilities for the seasonal prediction of meteorological drought. Journal of Applied Meteorology and Climatology, 51(7), 1222-1237.

McKee, T. B., Doesken, N. J., & Kleist, J. (1993). The relationship of drought frequency and duration to time scales. In Proceedings of the 8th Conference on Applied Climatology (Vol. 17, No. 22, pp. 179-183). Boston, MA: American Meteorological Society.

Rahman, M. M., Hagare, D., Maheshwari, B., & Dillon, P. (2015). Impacts of prolonged drought on salt accumulation in the root zone due to recycled water irrigation. Water, Air, & Soil Pollution, 226(4), 1-18.

Sheffield, J., Goteti, G., Wen, F., & Wood, E. F. (2004). A simulated soil moisture based drought analysis for the United States. Journal of Geophysical Research: Atmospheres, 109(D24).

Shokoohi, A., & Morovati, R. (2015). Basinwide comparison of RDI and SPI within an IWRM framework. Water Resources Management, 29(6), 2011-2026.

Sims, A. P., & Raman, S. (2002). Adopting drought indices for estimating soil moisture: A North Carolina case study. Geophysical Research Letters, 29(8).

Wilks, D. S. (2011). Statistical methods in the atmospheric sciences (Vol. 100). Academic press.

---

## Short Comment (SC3) · 31 Oct 2016

"Note to the editor and authors: As part of an introductory course to the Master programme Earth & Environment at Wageningen University, students get the assignment to review a scientific paper. Since several years, students have been reviewing papers that are in open online discussion for HESS, and they have been asked to submit their reports to the discussion in order to help the review process. While these reports are written as official reviews, they were not requested for by the editor, and we leave it up to the editor and authors to use these reports to their advantage. While several students were asked to review the same paper, this was not done to provide the authors with much extra work. We hope that these reports will positively contribute to the scientific discussion and to the quality of papers published in HESS. This report was supervised by dr. Ryan Teuling."

[Figure]

In this manuscript SPI and RDI are compared to soil moisture droughts using a physically based soil water model (Hydrus-1D) for three different climate zones. From this physically based soil water model monthly minimal values are compared to 3-monthly SPI and RDI. To calculate the Failure Rate (FR) and False Alarm Rate (FAR), thresholds for all values are set to the 75th percentile. The uncertainty of the model is taken into account by comparing perturbed input values to the original input values. If the FR and FAR for the perturbed values are higher than the original ones the simple drought index preforms better than the model. The FR an FAR for SPI and RDT ranged from 19% to 68%. There are three options stated why FR and FAR were not lower than 19%. It is concluded that the SPI preforms better than both the RDI and the model. However, to give a physically meaningful threshold it is advised to use a model over a drought index.

When reading this manuscript, I noticed several positive points. This research is daring, since comparing a soil water model to drought indices is not done often. The necessity for drought research is very clearly described in the introduction. The research done in this manuscript is line with the cope of this journal. However, there are some major issues that mostly involve the comparison of the meteorological drought indices to the model. Firstly, comparing an absolute value to an anomaly is not correct. Also, the thresholds taken to calculate FR and FAR are dubious. Finally, critical choices that have been made are not elaborated enough. These three points will be further explained in the next paragraphs. I also doubt the novelty of comparing SPI and RDI; these two drought indices are compared to each other multiple other papers already. Because the changes proposed will have an extensive effect on the methodology and thereby the results and possibly the conclusion, I would recommend a major revision.

The following arguments are specified in order of importance. 1) In step 3 of the method section it is stated that the minimum soil water pressure of each month is compared to SPI and standardized RDI. In my opinion comparing these two is fundamentally flawed. SPI and RDI are anomalies of a mean while the pF is taken as an absolute value. An

absolute value of pF does not say anything about how dry the soil is in comparison to normality. As it is used now, wet seasons would never have a drought according to the pF values, while the meteorological drought indices can point out a drought if there is less precipitation and more evaporation than normal. Therefore, a direct comparison of these two would be skewed, leading to misinterpretations of the correlation. If there is earlier research done saying that comparing the minimum pF value of a certain month to 3 monthly SPI or RDI is correct and how to interpret this result, I would like to have this explained and referenced to in the paper. However, in my opinion the comparison would be better if the pF is also transformed to a standardized index. I would propose to do this in the same manner as the SPI stated in McKee et al (1993). In that way all indices can be better compared to each other and relations found can be more easily interpreted.

2) The choice to take the 75th percentile as a threshold for the SPI and standardized RDI made in step 3 of the methods seems to be chosen arbitrary. How it is written now, the main reason seems that the values of FR and FAR would be the same if the 75th percentile is chosen as threshold. The threshold to define a drought occurrence for SPI and standardized RDI values is by definition 0. Every value below 0 indicates a drought. The table on page 2 of McKee et al. (1993) gives clear definitions of what values of SPI indicate certain types of droughts. In the paper of Tsakiris and Vangelis (2005) it is said that that for the standardized RDI the same thresholds can be taken. Therefore, I would suggest to take the threshold of 0 or one of the other thresholds stated in McKee et al. (1993). If the pF is calculated to an index in the same manner as SPI, the threshold would even count for this index. When this is done research question 1 (page 2 line 27) can be rephrased to: "Is it sufficient to use a simple drought index such as the SPI or RDI?".

3) Overall the choices made in the method seem arbitrary. There is little to no elaboration as to why critical choices have been made. The major issue that is not elaborated enough is the choice of using SPI and RDI and the Hydrus-1D model. In my opinion the

manuscript would be much stronger if more drought indices were taken into account. Given that the data is already available, this is relatively easily done. Therefore, I am curious to know why only these two drought indices are used. In the reference made for the Hydrus-1D model (Simunek et al., 2008), several other models for calculating soil moisture have been given, so it does not make clear why this particular model is chosen. In general, the method step 2 is not worked out enough. The reference period for the calculation of the non-exceedance probability of the SPI is not given. The calibration and validation period for the Hydrus-1D model is not given. In addition, there are too little references used in the methods from step 3 on, making the fundamentals of the research weak.

There are also more minor arguments to address before publishing this manuscript. 1) Other parts, more minor parts, of the methodology were not well elaborated. E.g., page 3 line 26. Why show the three-month averages and not the one or twelve-month instead? Or page 4 line 18. Why use 5 and 30 cm depth? Or page 6 line 10. Why use a range of -50% to + 50% for the perturbation?

2) Page 2 line 27-28. Research question 1 does not seem specific enough. If I understand correctly, it is meant as: Is it sufficient to use a simple drought index such as the SPI and RDI to estimate soil moisture deficits, (. . .)? If this assumption is correct, the conclusion does not answer this question. The conclusion answers the question: When comparing SPI and RDI, which one is better at detecting a soil water deficit?

3) Page 4 line 25. Assumption iii in the method section 2.2 states that "free drainage lower boundary condition is an adequate approximation". I doubt that this is the same for the three different climatic zones.

4) Page 10 line 18-19. Are the results and thereby the conclusions still reliable after stating that the outcome may be due to the accuracy of the model?

5) Page 9 line 1-2. Where is the phrase "a physically based soil water model should be used in preference" based on in this manuscript? I could not find evidence for this

statement in this research.

6) Page 3 line 2. The relevance to ecosystem restoration applications does not come back later in the paper. Please either leave this sentence out or refer to the possible applications of this research in the discussion.

7) Page 5 line 20. The reference used (Arnold et al., 2014) is very specific for the germination of one species. Stretching this to all phases of plant life and all plant types seems not right. A reference to a broader paper would be better.

8) Page 9 line 19-20. The research of Sims and Raman (2002) differs to much from this research to compare. Comparing to Khalili et al. (2011), Pashiardis & Michaelides (2008), or Zarch et al. (2015) would be more appropriate.

9) Page 1 line 31. The reference to Vicente-Serrano et al. (2010) does not seem a good reference for this sentence. Referring to Zagar et al. (2011) would be more appropriate.

Other minor issues are listed here in order of occurrence in the manuscript. Page 1 Title. There is no prediction it this paper. Please rephrase the title. Page 1 line 21. Change "provide physically" to "provides physically". Page 4 line 18. Change "soil depth" in "soil depths" and give the values instead of "(see below)" Page 4 line 22. Strange place of reference. Page 5 line 13. Please write "FAR" in italic like "FR" in line 11. Page 6 line 6. Check equation for parentheses. Page 6 line 25. Check equation on font size. Page 7 line 2. Change "Appendix B1" in "Appendix B2". Page 7 lines 9-14. Better to provide this information also in a table. Page 8 line 11-12. Please rephrase. Page 10 line 3-7. This might be better in the methods. Page 10 line 24. What actually implies that PET is more important for the shallower soils? Page 11 line 31. Please rephrase "unlikely to more useful" to "unlikely to be more useful". Table 1 and 2, and Appendix A. These do not seem very necessary. Table 2. Please explain symbols in the caption. Table 2. Please check superscript in row 2 and 3. Table 2. What is "10" doing under Bourke row 2? Table 3. Please change "soil water for" to "soil

water pressure for" in the caption. Table 4. I do not think this information is relevant for this paper. Figure 1. Block 1. Why state "25 years (1988-2013) (3 sites)" while on the Bourke site the data is from 1971-1996. Figure 1. Block 2. Change "Step 1" in "Step 2". Figure 4. In the text (page 7 line 12) it was said that some form of comparison would be shown between the sites. However, only the Bourke site is shown. Figure 4. Suggestion: draws the lines of the 75%tile threshold in this figure. Figure 5. Elaborate more on the middle plot, which one is the 5 and which one is the 30 cm soil depth. Now this plot is not clear and is better to be taken out. Figure 6. Why not take also parameter n for the calculation of FR*? All appendices. These could all be part of the regular tables and figures.

References Khalili, D., Farnoud, T., Jamshidi, H., Kamgar-Haghighi, A. A., & Zand-Parsa, S. (2011). Comparability analyses of the SPI and RDI meteorological drought indices in different climatic zones. Water resources management, 25(6), 1737-1757.

McKee, T. B., Doesken, N. J., & Kleist, J. (1993, January). The relationship of drought frequency and duration to time scales. In Proceedings of the 8th Conference on Applied Climatology (Vol. 17, No. 22, pp. 179-183). Boston, MA: American Meteorological Society.

Pashiardis, S., & Michaelides, S. (2008). Implementation of the Standardized Precipitation Index (SPI) and the Reconnaissance Drought Index (RDI) for regional drought assessment: a case study for Cyprus. European Water, 23(24), 57-65.

Sims, A. P., & Raman, S. (2002). Adopting drought indices for estimating soil moisture: A North Carolina case study. Geophysical Research Letters, 29(8).

Šimůnek, J., van Genuchten, M. T., & Šejna, M. (2008). Development and applications of the HYDRUS and STANMOD software packages and related codes. Vadose Zone Journal, 7(2), 587-600. Tsakiris, G., & Vangelis, H. (2005). Establishing a drought index incorporating evapotranspiration. European Water, 9(10), 3-11.

[Figure]

Zarch, M. A. A., Sivakumar, B., & Sharma, A. (2015). Droughts in a warming climate: a global assessment of Standardized precipitation index (SPI) and Reconnaissance drought index (RDI). Journal of Hydrology, 526, 183-195.

Zargar, A., Sadiq, R., Naser, B., & Khan, F. I. (2011). A review of drought indices. Environmental Reviews, 19(NA), 333-349.

---

## Referee Comment (RC3) · Dr. Teuling (Referee) · 3 Nov 2016

Over the past few week, I have supervised three students that reviewed this manuscript as part of their MSc programme. They have submitted their reports individually, but my own assessment largely overlaps with their assessments. Hence, I will not redo their work, and hope the editor will use their reports instead. In summary, I find the comparison between standardized drought indices (i.e. based on a seasonally varying threshold) and what can be referred to as dryness indices (i.e. pF) poorly motivated at best. Why should these two be correlated, and if they are, what is the reason for this? Based on the arguments provided by my students, I believe this manuscript needs major revisions, including at least new analysis on standardized pF values.

---

## Author Comment (AC2) · 13 Nov 2016

We would like to thank the Anonymous Referee #1 for this review and the constructive comments. We have addressed the referee's major comments (addressed earlier in the discussion stage) and the minor comments are as follows.

General comment

The paper tests the capability of simple meteorological drought indices to detect drought events, as defined by simulated soil moisture time-series. The topic is of interest for practical applications in drought monitoring, since simulations are often hard to be performed over some areas. My opinion is that the overall quality of the paper is negatively affected by some basic assumption made by the authors during the analysis, which are not clearly presented and sometime poorly described. Often the

reported results seem off, due to errors or unclear explanations. Hence, I suggest to the authors to carefully reread the paper before to proceed with a full evaluation of the paper.

Response: All the authors are confident that the paper is ready for review. We regret and partly disagree fundamentally with some of the referee's views. We hope that the addition of new text to the manuscript will address some of the issues associated with the analysis, which were not clear to the reviewer and/or required further information. We await with interest the comments of other reviewers.

Specific comments

Comment 1.1: First of all, they compared the 3-month SPI and RDI against a time-series of monthly minimum pF. If I have understood correctly, this means that this time series is obtained by choosing the minimum pF value (out of roughly 30 values) for each month in the simulation period. If this is the case, I'm really surprised to see the really good correspondence between SPI and pF as shown in Fig. 4 (and 5 as well). Since SPI (as well as RDI) is a standardized variables, its "random" behavior in Fig. 4 is justified, but the same cannot be said for minimum pF which should retain a sort of seasonality depending on the climate of the area. I'm not familiar with the climate of the specific study region, so it is possible that this behavior is due to the peculiar climate of the region, but in general it is advisable to perform a correlation analysis between a standardized variable (SPI) and a non-standardized one.

Response 1.1: SPI and RDI are not standardised seasonally in this application (Eqs. 1 and 2) and therefore they do include seasonal patterns. In this regard the good correspondence between the SPI and pF is expected. The correspondence and seasonality of the SPI and pF can be seen in the raw data plotted in figure 1. Originally we only plotted the simulated soil water pressure for Bourke, which has limited seasonality - where the ratio between winter and summer rainfall is 0.61 (Table 1 in manuscript). We have revised this figure to now include Melbourne and Cairns, where Cairns shows

much more distinct seasonal patterns (with a ratio of 0.10 ) so that the seasonality of the locations and the lack of standardisation is clearer (see revised Fig. 1 below). We strongly disagree that "in general it is not advisable to perform a correlation analysis between a standardized variable (SPI) and a non-standardized one". The definition of any index is rather arbitrary and usually specific to the pre-defined problem. That said, we believe it is essential to compare indices with physically measurable and plausible variables – no matter if the index includes a standardisation process or not. Ideally, indices such as the SPI or RDI are compared with empirical field data. However, such empirical data are often not available (such as in our study locations) for a variety of reasons, though primarily because long-term monitoring programs are restricted due to limited funding and time. The lack of empirical data is an issue across the world especially in developing nations or nations such as Australia with little history of long-term monitoring programs. Therefore, a logical step in absence of such data is to apply physically based models such as Hydrus-1D with available empirical data such as rainfall/evaporation and soil water retention characteristics. We believe our study addresses this critical step!

Comment 2: Also, the authors do not clarify if the 75% threshold is computed separately for each month or for the whole dataset. I assume is the first case (based on the data in Fig. B), but this is never clearly stated. Following this topic, in the same figure it seems that the 75% threshold corresponds to an SPI value around 1.2. This means either that: 1) both tails of the distribution are accounted in this computation, but the correct approach would be to consider just one tail since drought event (i.e., extreme dry conditions) are analyzed here, or 2) the fitting of you distribution is poor since the theory suggests that only about 11% of the data should be < -1.2 according to the normal distribution (about 3 values). It is fundamental that this issue is clarified and eventually fixed.

Response 2: The assumption that the 75% threshold is computed separately for each month is incorrect. In Section 2.3 (P5 L 17-18) we state that "For each site the threshold

that determines a soil moisture drought event was selected by the percentile of all simulated pF5 and pF30 . . ." Further the referee has assumed that the 75% threshold is computed separately for each month based on Figure B. We regret that the conceptual schematic in Fig. B may have caused a misunderstanding as what was shown in the diagram was example data. We will clarify that in a revised version of the manuscript. In this regard, the further comments of the referee under comment 3 are irrelevant.

Comment 3.1: The analysis on extreme values is really misleading, and it also needs to be extended by including other indices. The authors say that FR and FAR are identical in all the cases, but this shouldn't be the case. FR is equal to FAR only if a and c are the same, but this is really unlikely to happen in real cases. For instance, in your example in Fig. B (which I assume is from one of your cases): FR = 5/8=62.5% whereas FAR = 4/7 = 57.1%. Please recheck your calculation of those indices.

Response 3.1: The RDI was excluded from the analysis of extreme values because of the SPI performed better (Table 3). The RDI results may easily be added to the Appendices though upon further reviewers' comments. We regret that the reviewer has been misled by the unequal number of data points in Fig B (a+b and b+c). This will be addressed in a revised manuscript (see revised App. B (Fig. 2 below)).

Comment 3.2: Also, FR and FAR are not the only indices relevant in this case, e.g., what about the skill of the SPI? Is it better than the climatology or the random case?

Response 3.2: It is not clear to us what the reviewer suggests by the 'skill' of the SPI. It will be easy to add additional performance indices, although our current view is that the visual assessment, R2 values and FR/FAR are sufficient.

Comments 4.1: Finally, the results of the sensitivity analysis are surprising and need some clarifications. In almost all the case you have FR/FAR values between 30 and 50% higher than in the case of SPI. This means that FR/FAR values for the perturbed simulation are in the order of 65-70% in all cases, included several cases where only a 10% error in 1 parameter is added/subtracted (ie., Bourke 5 cm, Cairns 5 cm, Melbourne 5 cm). I'm really surprised by this result, since in my experience, even for a very sensitive parameter, a 10% change can rarely leads to have 2/3 of the previously detected extremes not detected anymore.

Response 4.1: The Richards' equation is used in the Hydrus model (P4 L24). Given the non-linearity of the water retention curve even a $\pm10\%$ changes in the van Genuchten parameters disproportionately affect the calculated values (Šimunek et al., 2012).

Comment 4.2: It would be useful to have a figure with the reference and perturbed simulations (only the maximum and median ones), as well as the corresponding threshold values, in order to better understand how these changes affect the results.

Response 4.2: We agree with the referee's suggestion, however we are concerned about the large number of graphs (3 sites, 2 depths, 3 model parameters = 18 graphs). Therefore as an example we propose to present one graph in Appendix D – see Figure 3 below.

Comment 4.3: Also, judging from Fig. B it seems that the same 75% threshold is used for both the reference and the perturbed simulation. I assumes that this is not the case, and it is just a coincidence, but I suggest to clarify in the text that the 75% threshold is adapted for each simulation accordingly to the simulated values.

Response 4.3: The reviewer's assumption is not correct. The threshold of the perturbed pF is same as the 75th percentile of the default pF, as shown in Figure 2 and stated in the caption. It would not be a useful performance analysis otherwise. We will emphasise this in the revised manuscript.

Minor comments

1. P1, L6. Replace evapotranspiration with potential evapotranspiration:

We will replace evapotranspiration with potential evapotranspiration

2. P1, L6. Rephrase as "used as proxy of severity and duration: : :." :

[Figure]

We will rephrase as suggested.

3. P1, L15. ": : :the frequency with which the simulated: : : below threshold". Actually, you do not want to estimate the frequency, since the frequency is already known as soon as the threshold is defined. Please rephrase.

Rephrased to: ". . . and their capability to detect soil moisture droughts potentially critical for plant water stress"

4. P2, L7-8. ". . .water is controlling. . .. (e.g., water cycle)". Please rephrase.

Rephrased to: ". . . as it is the key variable controlling many processes in biogeochemical cycles such as . . ."

5. P3, L6. I would rephrase as something like "The analysis . . ." Since you are not actually strictly describing a "method".

We will rephrase method to analysis

6. P3, L 27. How many years were used for the fitting? Which period (the full period?). Please clarify. Also, you should say something about the quality of the fittings (the same is true for RDI).

For both SPI and RDI we used the total length of rainfall/evaporation (table 1) and the quality of the fitting was always $R2>0.97$ and RSME = 0.016-0.052

7. P4, L21. Appendix A is just a table. Do you really need an appendix for a table? Same for the other appendices.

In order to maintain the flow and the consistency of the manuscript, we prefer keeping the Hydrus-1D configuration in the appendix.

8. P4, L22. Remove the parenthesis before "Australian" and move it before "2011". Thanks for pointing this out. We will make the change.

9. P5, L1. Please clarify if minimum means minimum among the 30sh daily values in a

specific month. Also, please include a standardization of the variable for the successive comparisons with SPI, RDI.

It's the monthly minimum for both 5 cm and 30 cm soil depths. Detailed explanation of standardization of pF values are provided under responses of reviewer #3

10. P5, L8-16. Please add at least a skill score.

Addressed under major comment 3.2

11. P5, L17-24. This part on the definition of the threshold is unclear. Please clarify if the threshold is calibrated or not, since you contradict yourself successively in the text. Also, is the threshold computed for each month separately (e.g., 12 thresholds) or for the whole year? The first would be definitely better for the pF.

Addressed under the responses of major comments 2.

12. P5, L20. In Arnold et al. (2014) is reported that there is still seeding also at the wilting point, which does not means that there is no stress. The capability to germinate is clearly reduced compared to optimal water conditions. You should check Cammalleri et al. (2016) "A novel soil moisture based drought severity index (DSI) combining water deficit magnitude and frequency" where a combination of water stress and frequency is used to define drought from simulated soil moisture. Your definition based only on frequency can lead to erroneous estimates over wet areas.

We will discuss the reference in the light of our approach and findings. Interestingly, Cammalleri et al. use a complex land-surface model to infer long-term soil moisture data, demonstrating the limited availability of long-term empirical soil moisture data to test the capability of drought indices to detect soil moisture droughts. This encourages us that our approach of using empirically derived soil water retention curves and a physically based soil water model as a reference or control scenarios is a valuable alternative (Response 1.1).

13. P5, L21. "all values below zero: : :". It seems that this is not what was done since

the

75% threshold is not at zero. We agree with the reviewer and will remove that sentence.

14. P5, L22-24. This statement is true only in theory (see Fig. B) and it also highlights how it does not make sense to test two indices if is known a-priori that they would be the same.

Addressed under major comments 2 and 3.1

15. P6, L9. How do you define the "most extreme droughts"? How was the interval -0.5, 0.5 chosen? Also, please report that 10% steps were adopted.

Droughts are $\leq$ -2 considered as extreme droughts (McKee et al., 1993) we will include the reference to the text.L10. We selected $\pm 0.5$ assuming the input parameters may vary between this ranges. We will include in the method that we used $\pm 10\%$ steps.

16. P6, L9-14. To compare extreme pF values in different sites does not make much sense, since one site can be "naturally" drier than another which is not related to the occurrence of a drought event (e.g., pF in a dry area after a rainy period can be higher that pF during a drought in Sweden). This is the reason why standardized SPI is used.

Our underlying assumption is that native plants have been established over long periods and are adapted to the local environmental condition and would suffer similar water stress at the 75th percentile soil water pressure across the three locations. Of course this implies different absolute quantities of soil water pressure. For example, the 75th percentile corresponds to pF 3.4 in Bourke, but is only pF 2.3 and 2.1 in Melbourne and Cairns, respectively (Fig. 5 in the manuscript).

17. P6, L28-29. This contradict what stated in the methodology.

P 6, L28-29 is under methodology

18. P8, L28-30. The FR alone cannot fully explain the performance of SPI. For instance, is this better that randomly guessing drought events? E.g, How skillful is this

index?

Addressed under major comments 3.2

19. P9, L20. Is more representative compared to what? (I assume to PDSI considering the reference). I would rephrase as "it represent well . . .".

It has to be changed as "SPI is representative of soil moisture variation"

20. P10, L9. "rather well: : : then: : :". Please rephrase.

" . . . than for deeper soils."

21. P10, 18-19. This sentence is not clear; also, Fig. 5 seems not relevant to this discussion. : A particularly interesting result was that the SPI index, which excludes the effect of PET, performed considerable better than the RDI index (Fig. 5), which may have been due at least partly to the accuracy with which our application of Hydrus simulated evaporation."

The web plot in the centre of Fig. 5 illustrates that the correlation between RDI and pF greater than SPI vs pF in Bourke.

22. P11, L11-12. This is not necessarily the case. If a significant trend in soil moisture is observed on the site, the use of a longer time-series could negatively affect the analysis (without proper de-trending, etc.).

We will consider this as a limitation in the revised manuscript. Thank you.

23. Fig. 4. Please report the starting/ending dates, as well as a time scale that is multiple of 1 year (e.g., 12-24, 48: : :) to make the figure more readable.

We will revise the figure 4 aligned with the editors suggestions and will include further details as necessary.

24. Fig. 5. "The plots represent the highest correlation". What does it means? Please clarify.

At each location we have assessed 4 perturbations of 2 soil depths and 2 indices. The scatter plots represent the highest correlations between these two variables.

25. Fig. 6. Please re-arrange this figure to make it clearer. E.g., order for site and depth, etc.. Also, the acronym C30, M5, etc. are not defined. Please clarify.

Bars in the figure 6 arranged from highest failure rate to the lowest "The difference between the FR*-FR for all sites and soil profiles for SPI". The C, M, B, are referred to Cairns, Melbourne, Bourke and 5 and 30 refers to the soil depth. We will define them in the figure caption.

References

McKee, T.B., Doesken N. J, Kleist John, 1993. The relationship of drought frequency and duration to time scales, Proceedings of the 8th Conference on Applied Climatology. American Meteorological Society Boston, MA: Anaheim, California, pp. 179-183. Šimunek, J., Van Genuchten, M.T., Šejna, M., 2012. HYDRUS: Model use, calibration, and validation. Transactions of the ASABE 55(4) 1263-1274.

Please also note the supplement to this comment:
http://www.hydrol-earth-syst-sci-discuss.net/hess-2016-467/hess-2016-467-AC2-supplement.pdf

[Figure]

**Fig. 1.** Simulated monthly minimum soil water pressure in 5 cm depth and SPI for Cairns, Bourke and Melbourne.

[Figure]

Figure 2: (a) Conceptual schematic of periods of the simulated soil water pressures in relation to the calculated drought index. The threshold values (dashed lines) divide the schematic into four segments, upon which the failure rate (FR) and false alarm rate (FAR) are based on (Section 2, step 3). The segments represent the simulated low soil water pressure events that are not (a) or are (b) detected by the drought index, and the drought events detected by the drought index that do not (c) or do (not labelled) correspond to periods of low simulated soil water pressure. (b): Conceptual schematic of periods of the perturbed Hydrus parameters in relation to the default Hydrus parameters. The threshold values (dashed lines) for default is 75th percentile and the threshold value for perturbed is the soil water pressure value of 75th percentile of default (y).
Note, this is a schematic and the data do not represent the results of this study

**Fig. 2.**
Interactive
comment

[Figure]

**Fig. 3.** Default and perturbed (median and maximums for parameter alpha) monthly minimum soil water pressure in 5 cm depth for Bourke.

---

## Author Comment (AC3) · 13 Nov 2016

We would like to thank the Anonymous Referee #2 for this review and the constructive comments. We appreciate the reviewers' positive comments on the significance of our work "in agriculture drought monitoring in places without adequate soil moisture observations". We have addressed the referee's comments as follows:

Comment 1: As the key idea about the research is testing the ability of meteorological drought indices in predicting soil drought. Why only use soil water pressure to quantify the 'soil moisture droughts'? I suggest the author should use the observed soil moisture to test the capability of these drought indices. You may not use the SM data of all layers studied, at least the average condition of SM and its correlation with the drought indices should be revealed. Comment 2: As agricultural drought or eco-drought are

usually measured by soil moisture. The relationship between soil moisture and soil water pressure used in current research should be further studied in the 3 stations.

Response 1 & 2: We use soil water pressure over soil moisture as this allows us to examine potential water fluxes between plant roots and the soil water storage. Only water pressure provides the relevant information to assess water availability for plants, which is required for estimating agricultural droughts. The unique relationship between soil water pressure and soil moisture is described by the water retention curve (Table 2 in manuscript). Implicitly, water retention characteristics are usually fundamental to soil moisture based indices such as the soil water deficit index (Martínez-Fernández et al., 2015). As the soil water retention curve is monotonic, we have no reason to believe that the use of soil moisture over soil water pressure would affect our findings on the false alarm (FAR) and failure rate (FR). However, we acknowledge the role of soil moisture defining the total store of water and provide the web plots of correlations between soil moisture and drought indices (Fig. 1 below), as well as the corresponding simulated time series (Fig. 2 below). We do not propose to include these Figures in the paper, but we thought useful to show these for the review process.

Comment 3: I suggest the author also analyze the effect of drought timescale on soil moisture. You may analyze more on soil moisture and drought with changing timescales e.g.1-12months.

Response 3: "The SPI and RDI were calculated using climate inputs averaged over one, three and twelve month time periods. However for the sake of simplicity and to keep the paper length reasonable we present only the results based on the three month time period" (manuscript P3L25). The correlation was most significant at the 3-monthly timescale and qualitatively the same across all timescales (Figure 3, not included in paper).

Comment 4: In addition to model parameter setting, the input of the model including the climatic data should be clarified to enhance the comparison between model output

and the drought indices calculated from precipitation/PET.

Response 4: Without further details we don't know what further information is requested by the referee other than Tables 1 and 2 in the manuscript.

Comment 5: In discussion, the author mentioned that 'our results point to the simplest being the best'. Such kind of expression should be very careful as the study only analyses SPI and RDI. Actually there are many effective drought indices with precipitation and PET, e.g. SPEI. The author can read more literatures on this.

Response 5: We agree with the reviewer here and will expand our discussion on the capability of simple drought indices to detect soil moisture droughts and explicitly consider the references provided by Reviewers #1 and #3.

References

Martínez-Fernández, J., González-Zamora, A., Sánchez, N., Gumuzzio, A., 2015. A soil water based index as a suitable agricultural drought indicator. Journal of Hydrology 522 265-273.

Please also note the supplement to this comment:
http://www.hydrol-earth-syst-sci-discuss.net/hess-2016-467/hess-2016-467-AC3-supplement.pdf

———————————————————

**Fig. 1.** Correlations between simulated monthly minimum soil moisture in 5 cm soil depth vs SPI and RDI. The scatter plots represent the correlation for each location.

[Figure]

Figure 2.1: Time series of the SPI, the simulated monthly minimum soil water pressure and monthly average soil water pressure, and the monthly minimum soil moisture and monthly average soil moisture in 5 cm depth in Cairns. Note: average and minimum soil moistures are also included to this figure aligned with the reviewer #3 comments.

**Fig. 2.**

[Figure]

Figure 2.2: Time series of the SPI, the simulated monthly minimum soil water pressure and monthly average soil water pressure, and the monthly minimum soil moisture and monthly average soil moisture in 5 cm depth in Bourke. Note: average and minimum soil moistures are also included to this figure aligned with the reviewer #3 comments.

**Fig. 3.**

Figure 2.3: Time series of the SPI, the simulated monthly minimum soil water pressure and monthly average soil water pressure, and the monthly minimum soil moisture and monthly average soil moisture in 5 cm depth in Melbourne. Note: average and minimum soil moistures are also included to this figure aligned with the reviewer #3 comments.

**Fig. 4.**

**Fig. 5.** Correlations between simulated monthly minimum soil water pressure (pF) vs SPI and RDI for 5 cm and 30 cm soil depth at a 1-, 3- and 12-monthly timescale, respectively.

---

## Author Comment (AC4) · 13 Nov 2016

We would like to address the major issues identified by all the reviewers (short communication #1, #2 and #3) in a compiled response (under the response for reviewer #3), and address the minor comments in a revised version of the manuscript. We are confident that the reports will make a positive contribution to the quality of our manuscript.

---

## Author Comment (AC5) · 13 Nov 2016

We would like to thank Judith Poelman for the constructive comments. We will address the major issues identified by all the reviewers (short communication #1, #2 and #3) in a compiled response (under the response for reviewer #3), and address the minor comments in a revised version of the manuscript. We are confident that the reports will make a positive contribution to the quality of our manuscript.

---

## Author Comment (AC7) · 13 Nov 2016

We would like to thank Dr Ryan Teuling and his students Danny Heuvelink, Judith Poelman and Heleen Westerveld for their critical and constructive comments on our manuscript. We appreciate the time and effort taken to review our manuscript. We would like to address the major issues identified by all the reviewers in a compiled response, and address the minor comments in a revised version of the manuscript. We are confident that the reports will make a positive contribution to the quality of our manuscript.

All three reports emphasise the relevance and novelty of our topic for the research community as well as practical applications and the fit within the aims and scope of HESS. While the introduction is deemed to be well written, all reviewers stress the

lack of explanation around some fundamental assumptions made in the methods that would certainly help to better understand the implications of our findings. Specifically, the reviewers' concerns are related to (1) the use of the minimum over the average monthly soil water pressure as reference for potential plant water stress, (2) the use of SPI/RDI and Hydrus-1D over other indices or numerical soil water models, and (3) the comparison of standardised indices with simulations of a physically based model or empirical measurements (including the use of the 75th percentile threshold). We note that concerns (2) and (3) are in line with that of Reviewer #1 and would like to further expand on our earlier response to Reviewer #1 below.

(1) Use of the minimum over the average monthly soil water pressure as reference for potential plant water stress

The decision to use the minimum rather than average monthly soil water pressure as reference for potential plant water stress is based on the assumption that one incidence of exceeding a species-specific water pressure threshold causes irreversible plant water stress. This reference point is more biologically relevant than average monthly soil water pressure as averages may mask high variability. In this regard, we make a very strong assumption about the (lack of) mechanisms plants may have developed to overcome short periods of water stress. We acknowledge the alternative assumption, in which case the average monthly soil water pressure would provide the better metric for comparing between SPI/RDI and Hydrus-1D. We will address this in a revised version of the manuscript by discussing the results obtained using the alternative approach.

Please find below plots of the two alternative metrics for the three locations in our study (Figs. 1.1 – 1.3), as well as the web plot of correlations between the indices and the average monthly soil water pressure (Figure 2). Though there is a good qualitative correlation between monthly average and monthly minimum soil water pressure (Figure 1.1-1.3) the correlation values between drought index and average soil water pressure are always lower than the monthly minimum soil water potential except for Melbourne (compare Fig. 2 below and Fig. 5 in the manuscript). Further, there is no

interesting/significant trend or variation between the monthly average soil water pressure with two soil depths and two drought indices compared to monthly minimum soil water pressure.

(2) Use of SPI / RDI and Hydrus-1D over other indices / numerical soil water models

The objective of our study is to test the capability of a simple meteorological drought index to detect relevant periods of deficits in soil water availability. Note that the objective was not to compare the performance of drought indices amongst each other or to compare alternative soil water models. We realise that, in order to make this more explicit in the manuscript, we have to carefully rephrase parts of the introduction and methods.

In order to meet this objective, we selected the SPI as a representative index out of the great pool of meteorological drought indices from the literature as it considers rainfall as the only input variable. Also the SPI is one of the most commonly used indices and tends to be used for more than just meteorological droughts in practice. Acknowledging the critical impact of evaporation on the soil water balance we selected the RDI as an alternative simple drought index using evaporation as an additional input variable to rainfall. Any additional input variables or the use of a generic two-layer soil model (as in the PDSI proposed by one reviewer) would compromise our objective and be out of scope.

In regards to the numerical soil water model, we selected Hydrus-1D because it is a well-established soil water flow model that is freely available, which ensures the reproducibility of our work. We acknowledge that the model selection is a somewhat random process. However, any numerical model is, to some degree, a simple representation of physical processes and would have limited predictive power. The uncertainty in the model is addressed using parameter sensitivity analysis, as is common practice in the environmental modelling literature. Ideally, indices such as the SPI or RDI are compared with empirical field data. However, such empirical data are often not available

(such as in our study locations) for a variety of reasons, though primarily because long-term monitoring programs are restricted due to limited funding and time. The lack of empirical data is an issue across the world especially in developing nations or nations such as Australia with little history of long-term monitoring programs. A logical step before implementing any long-term campaigns is to test their feasibility in a desktop study using physically based models such as Hydrus-1D with available empirical data such as rainfall/evaporation and soil water retention characteristics, as demonstrated in our work. That said, the model is used as a reference/control similarly as any empirical soil moisture data would be used and, hence, calibration/validation with empirical data (as requested by one reviewer) is deemed to be redundant (otherwise we would have used the empirical data as a reference in the first place).

(3) Comparison of standardised indices with simulations of a physically based model or empirical measurements (including the use of the 75th percentile threshold)

The main concern of all reviewers is that the indices are standardised quantities whereas the modelled soil water pressure is an absolute physically relevant metric. As emphasised in our response to Reviewer #1, the SPI is standardised using the long-term average rather than the seasonal averages. Therefore, the standardisation in SPI rescales the data and does not remove seasonal variability, so it is not expected to make much difference to the correlations (only insofar as changing from a skewed to a normalised distribution of values) and cannot affect the FR/FAR values (as the scaling is monotonic). We will emphasise this fact in the revised paper. The use of non-seasonally-standardised indices is not uncommon, (e.g. Martínez-Fernández et al., 2015; Wang et al., 2015; Wang et al., 2016)

Regarding the use of the 75th percentile, our underlying assumption is that native plants have been established over long periods and are adapted to the local environmental condition and would suffer similar levels of water stress at the 75th percentile soil water pressure across the three locations. Of course this implies different absolute quantities of soil water pressure. For example, the 75th percentile corresponds to pF

3.4 in Bourke, but is only pF 2.3 and 2.1 in Melbourne and Cairns, respectively (Fig. 5 in the manuscript). In order to address the issue of an arbitrarily selected threshold, we tested our methods within the range of 45-95% (Fig. 3 in the manuscript).

Standardisation and/or normalisation of soil water pressure (be it modelled or measured) would require further assumptions of which we have already made a lot (as pointed out by some reviewers). For example, a distribution function would be required for the standardisation process, which involves further uncertainty. Likewise, in the normalisation process the scale of the normalized interval is significantly affected by any outliers.

For the Reviewers' and Editor's information, we have transformed the modelled pF based on the mean and standard deviation of a normal distribution (Fig. 3). Yet the strength of the relationship remains the same as for the correlation presented in the manuscript (compare Fig. 4 below and Fig. 5 in the manuscript).

At the Editor's discretion, we would prefer to keep the study reasonably simple rather than adding further arbitrary transformations of the data and hope our findings will be considered as useful desktop study to justify further work on the capability of simple drought indices to detect plant water stress related soil moisture deficits, including the establishment of long-term monitoring networks for the verification/falsification of our findings.

Further comments that will be addressed in the revised manuscript:

J. Poelman, comment 2.1: Emphasise the effort made in former studies and further stress the novelty of our study in the introduction. J. Poelman, comment 2.4: Further references to justify the step-by-step description of methods. J. Poelman, comment 3: Expand discussion on when to use the indices over the model in relation to uncertainty in the water retention curves. All reviewers: All minor comments/issues, including references. Thank you!

References

Martínez-Fernández, J., González-Zamora, A., Sánchez, N., Gumuzzio, A., 2015. A soil water based index as a suitable agricultural drought indicator. Journal of Hydrology 522 265-273. Wang, H., Rogers, J.C., Munroe, D.K., 2015. Commonly Used Drought Indices as Indicators of Soil Moisture in China. Journal of Hydrometeorology 16(3) 1397-1408. Wang, H., Vicente-serrano, S.M., Tao, F., Zhang, X., Wang, P., Zhang, C., Chen, Y., Zhu, D., Kenawy, A.E., 2016. Monitoring winter wheat drought threat in Northern China using multiple climate-based drought indices and soil moisture during 2000–2013. Agricultural and Forest Meteorology 228–229 1-12.

Please also note the supplement to this comment:
http://www.hydrol-earth-syst-sci-discuss.net/hess-2016-467/hess-2016-467-AC7-supplement.pdf

[Figure]

Figure 1.1: Time series of the SPI, the simulated monthly minimum soil water pressure and monthly average soil water pressure, and the monthly minimum soil moisture and monthly average soil moisture in 5 cm depth in Cairns. Note: average and minimum soil moistures are also included to this figure aligned with the reviewer #3 comments.

**Fig. 1.**

[Figure]

Figure 1.2: Time series of the SPI, the simulated monthly minimum soil water pressure and monthly average soil water pressure, and the monthly minimum soil moisture and monthly average soil moisture in 5 cm depth in Bourke. Note: average and minimum soil moistures are also included to this figure aligned with the reviewer #3 comments.

**Fig. 2.**

Figure 1.3: Time series of the SPI, the simulated monthly minimum soil water pressure and monthly average soil water pressure, and the monthly minimum soil moisture and monthly average soil moisture in 5 cm depth in Melbourne. Note: average and minimum soil moistures are also included to this figure aligned with the reviewer #3 comments.

**Fig. 3.**

Figure 2: Correlations between simulated monthly average soil water pressure pF Vs three month time scale of SPI and RDI for 5 cm and 30 cm soil depth. The scatter plots represent the highest correlation for each location

**Fig. 4.**

[Figure]

**Fig. 5.** Figure 3: Standardised monthly minimum soil water pressure in 5 cm depth and SPI for Cairns, Bourke and Melbourne.

[Figure]

Figure 4: Correlations between standardized monthly minimum soil water pressure STD pF Vs three month time scale of SPI and RDI for 5 cm and 30 cm soil depth. The scatter plots represent the highest correlation for each location

**Fig. 6.**